# Engineering of extracellular vesicles for efficient intracellular delivery of multimodal therapeutics including genome editors

Xiuming Liang [1,2,3,4] ✉, Dhanu Gupta [1,5,6,25], Junhua Xie[7,8,25], Elien Van Wonterghem[7,8], Lien Van Hoecke[7,8], Justin Hean[9], Zheyu Niu[1,10], Marziyeh Ghaeidamini[11], Oscar P. B. Wiklander [1,2,12], Wenyi Zheng [1,2,3], Rim Jawad Wiklander[1], Rui He[13], Doste R. Mamand[1,2,12], Jeremy Bost [1], Guannan Zhou[1,14], Houze Zhou[1,2,3], Samantha Roudi [1,2,3], H. Yesid Estupiñán [1,2,3,15], Julia Rädler [1,2,3], Antje M. Zickler [1,2,3], André Görgens [1,2,3,16], Vicky W. Q. Hou[1,2,3], Radka Slovak [9], Daniel W. Hagey [1,2,3], Olivier G. de Jong [17], Aileen Geobee Uy[1,18], Yuanyuan Zong[19], Imre Mäger [20], Carla Martin Perez[5], Thomas C. Roberts [5,6,21], Dave Carter[9], Pieter Vader [22,23], Elin K. Esbjörner [11], Antonin de Fougerolles[9], Matthew J. A. Wood[5,6,21], Roosmarijn E. Vandenbroucke [7,8], Joel Z. Nordin [1,2,24] ✉ & Samir EL Andaloussi [1,2,3] ✉

Intracellular delivery of protein and RNA therapeutics represents a major challenge. Here, we develop highly potent engineered extracellular vesicles (EVs) by incorporating bio-inspired attributes required for effective delivery. These comprise an engineered mini-intein protein with self-cleavage activity for active cargo loading and release, and fusogenic VSV-G protein for endosomal escape. Combining these components allows high efficiency recombination and genome editing in vitro following EV-mediated delivery of Cre recombinase and Cas9/sgRNA RNP cargoes, respectively. In vivo, infusion of a single dose Cre loaded EVs into the lateral ventricle in brain of Cre-LoxP R26-LSL-tdTomato reporter mice results in greater than 40% and 30% recombined cells in hippocampus and cortex respectively. In addition, we demonstrate therapeutic potential of this platform by showing inhibition of LPS-induced systemic inflammation via delivery of a super-repressor of NF-κB activity. Our data establish these engineered EVs as a platform for effective delivery of multimodal therapeutic cargoes, including for efficient genome editing.

Protein-based therapeutics have a unique potential for the treatment of diseases. Whereas there has been great success developing therapeutic proteins against extracellular targets, of which a broad range have been approved for clinical application[1,2], intracellular delivery of proteins and RNA remains challenging due to the inherent impermeability of the plasma membrane[3,4]. Thus, numerous strategies have been developed to facilitate intracellular protein delivery. For instance, the iTOP system exploits NaCl-mediated hyperosmolarity and transduction compounds to achieve high efficiency delivery of proteins into primary cells, but this has limited in vivo potential[5]. Although Cell-penetrating peptides

(CPPs) have likewise shown promise in some applications, the potential limitations related to endosomal entrapment and toxicity have been reported[6–8]. Finally, various nanocarriers, such as lipid nanoparticles and polymers, are frequently utilized for intracellular delivery of proteins, but with similar drawbacks to CPPs[9–13].

Another drawback of these strategies is that their synthetic nature may cause various side effects when used in vivo. Thus, one strategy to overcome these issues is to harness natural delivery vehicles—extracellular vesicles (EVs). EVs are lipid bilayer enclosed particles that are secreted and taken up by all cell types to mediate intercellular trafficking of biologically active molecules[14–16]. However, there are presently two main challenges that must be resolved to achieve efficient intracellular delivery of proteins by EVs: i) enrichment of therapeutic proteins inside EVs in a soluble (or non-tethered), active form; and ii) efficient endosomal escape of therapeutic proteins into the cytosol of the target cell.

The enrichment of specific proteins inside of EVs has previously been achieved by fusing the target proteins to the cytoplasmic domain of EV-sorting proteins, such as CD63[17]. However, as these proteins remain bound to the EV membrane, this approach is not suitable for cytosolic delivery of cytoplasmic or nuclear proteins. To address this limitation, several technologies introducing cleavable linker peptides between the target protein and EV-sorting domains, and the EXPLORs system have been developed[18–20]. Although several of the above technologies have achieved intracellular protein delivery by modifying EVs, these solutions were either dependent on extracellular conditions or required multiple components to be co-expressed. In the present study, we exploit the engineered self-cleaving mini-intein (intein) derived from *Mycobacterium tuberculosis (Mtu) recA*[21], to connect the EV-sorting protein with the cargo protein, facilitating liberation of soluble cargo protein from the EV-sorting protein inside the EV lumen. This intein comprises the first 110 and last 58 amino acids of the 441 amino acid *Mtu recA* intein, with four additional mutations (C1A, D24G, V67L, and D422G) introduced to enable C-terminal cleavage in a pH-sensitive manner at 37 °C[21].

Since endocytosis is the most common mode of biomacromolecular uptake by cells, endosomal entrapment constitutes the primary barrier to the functional intracellular delivery of both therapeutic proteins and EVs[22,23]. Interestingly, fusogenic proteins derived from viruses have been found to mediate endosomal escape and facilitate the release of EV cargo into the cell cytosol[24,25]. In this study, we make use of the fusogenic protein, vesicular stomatitis virus G glycoprotein (VSV-G), as both an efficient endosomal escape activator and EV-sorting protein. Here, we have developed two systems that solve the aforementioned problems above in tandem to achieve high efficiency intracellular protein delivery harnessing engineered EVs (Fig. 1a). Altogether, this work demonstrates the great potential for EV-based therapeutic protein delivery.

## Results

### Development of the VEDIC system for high efficiency intracellular protein delivery by EVs

Traffic Light (TL) fluorescent Cre reporter cells were exploited to assess the potential of EVs for intracellular delivery of functional proteins (Fig. 1b)[26]. LoxP recombination by Cre results in the excision of the red fluorescent protein (RFP) DsRed, which subsequently leads to permanent expression of green fluorescent protein (GFP) (Fig. 1b). EVs, isolated by tangential flow filtration (TFF), derived from Cre alone, CD63-Cre (Cre fused to CD63 to enhance EV-enrichment), CD63-Intein-Cre (self-cleaving protein intein was introduced between CD63 and Cre to liberate Cre from CD63 inside the EV lumen)[21] or Intein-Cre (no EV-sorting domain) overexpressing cells could not achieve any recombination at any tested dose in reporter cells analyzed by flow cytometry (Fig. 1c-e and Supplementary Fig. 1a, c, d). Next, a comprehensive screen of 40 human- and 2 virus-derived fusogenic proteins to identify

efficient candidates was performed since fusogenic proteins were reported to enhance EV cargo endosomal escape[27]. CD63-Intein-Cre was co-transfected with the fusogenic proteins and EVs were subsequently isolated. Based on Nanoparticle Tracking Assay (NTA) particle counts, the same number of particles were added to recipient Cre reporter cells. While none of the human-derived fusogenic proteins induced Cre-mediated recombination, the fusogenic viral protein VSV-G significantly boosted Cre delivery after co-transfection with CD63-Intein-Cre, such that 66% and 98% of HeLa-TL and T47D-TL cells, respectively, expressed GFP two days after treatment with EVs (Fig. 1f and Supplementary Fig. 1b). Subsequently, VSV-G was co-expressed with the construct tested in Fig. 1c and e, and isolated EVs were incubated with reporter cells for 48 hours (h) and the recombination was assessed by microscopy. Of the conditions tested, only VSV-G plus CD63-Intein-Cre EVs achieved significant activity in reporter cells according to flow cytometry analysis (Fig. 1g), indicating that the EV-sorting domain (CD63), self-cleaving protein (intein) and endosomal escape booster (VSV-G) were indispensable for efficient intracellular delivery of Cre by engineered EVs. We term this approach the VEDIC (VSV-G plus EV-Sorting Domain-Intein-Cargo) system.

Next, different doses of the VEDIC EVs were added to reporter cells. We observed a clear pattern of dose-dependent recombination in B16F10-TL, HeLa-TL and T47D-TL cells (Fig. 1h, i, Supplementary Fig. 1e-f and Supplementary Fig. 2a, e). For VEDIC EVs, a time-lapse video obtained from live imaging of HeLa-TL recipient cells showed increasing recombination over time (Supplementary Movie #1), while no GFP signal was observed in the absence of VSV-G (Supplementary Movie #2). Furthermore, the protein level of Cre and VSV-G in T47D-TL and B16F10-TL cells was evaluated and correlated with the GFP signal in these cell lines (Fig. 1j and Supplementary Fig. 2c). Cre could also be detected at lower levels in the non-VSVG group, indicative of cellular uptake but most likely entrapment in the endosomes (Fig. 1j and Supplementary Fig. 2c). Moreover, VEDIC EVs were applied to multiple hard-to-transfect cell lines, and we observed significant recombination 2 days after EV addition in all cell lines (Supplementary Fig. 1g-j and Supplementary Fig. 2b). Finally, CD63 was replaced with other known EV-sorting domains (CD81, CD9 and PTGFRN) and we found that addition of EVs isolated from cells transfected with these additional constructs resulted in significant recombination when combined with VSV-G (Supplementary Fig. 2d).

In order to know if the EV isolation or purification methods affect the potency of VEDIC EVs, we further purified the EVs after TFF by using Size Exclusion Chromatography (SEC) to evaluate their functionality in reporter cells (Supplementary Fig. 2f). As shown in Supplementary Fig. 2g, the potency of purified VEDIC EVs was preserved in the tested reporter cells.

To demonstate the utility of VEDIC EVs, we compared them with the published Nanoblade VLP system to deliver Cre recombinase and found VEDIC system was as good as Nanoblade in T47D-TL cells, but showed better activity in HeLa-TL and B16F10-TL cells (Fig. 1k)

In addition to adding engineered EVs to reporter cells, the EV-mediated recombination efficiency in co-culture assays was evaluated using direct co-culture (co-culture), IBIDI co-culture μ-slide (IBIDI) and Transwell (Transwell) assays (Supplementary Fig. 3)[28]. For direct co-culture assays, EV-producing and reporter cells were cultured at different ratios and GFP levels were evaluated one day after incubation (Supplementary Fig. 3a). Here, VEDIC expressing cells enabled significant recombination in various reporter cells (Supplementary Fig. 3b-e). For IBIDI co-culture assays, reporter cells were seeded in the inner reservoir as shown in Supplementary Fig. 3f, and the feeder cells were seeded in the 8 surrounding reservoirs. After 4 days of incubation, co-culture with the VEDIC donor cells showed significant levels of GFP activation in the reporter cells (Supplementary Fig. 3g, j). For Transwell assays, the reporter cells were seeded in the lower compartment and the EV-producing cells in the upper chamber

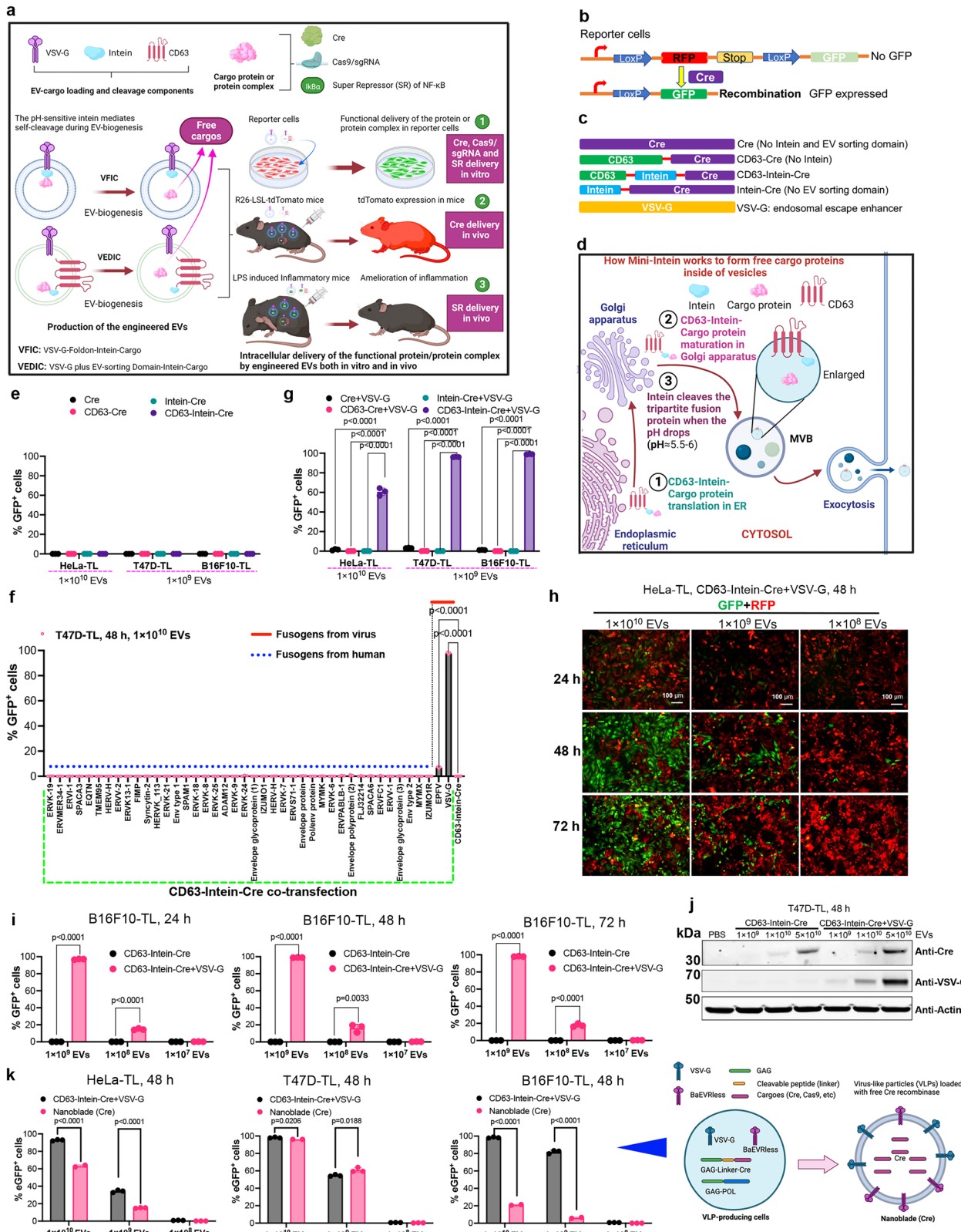

(Supplementary Fig. 3f). In line with the previous results, significant increases in the number of GFP positive cells was only observed after co-culture with the VEDIC donor cells (Supplementary Fig. 3h, i, k).

### Development of the VFIC system to further improve intracellular protein delivery by EVs

To further improve the VEDIC system, which requires transfection of several plasmids, we aimed to combine the essential components for intracellular EV cargo delivery into a single construct. To determine whether VSV-G itself could be employed as an efficient EV loading protein, VSV-G-mNG was transfected into HEK293T cells and the isolated vesicles were analyzed by single vesicle flow cytometry[29]. The total number of mNG+ vesicles was far greater than CD63+ vesicles (Fig. 2a). Upon VSV-G-mNG co-transfection with CD63 plasmid in HEK293T cells, the total number of isolated mNG+ vesicles were similar to that of CD63+ vesicles (Fig. 2a). These data imply that VSV-G could

**Fig. 1 | Development of the VEDIC system for high efficiency intracellular protein delivery by EVs. a** Graphic abstract of the development of VEDIC and VFIC systems for high efficiency intracellular protein delivery in vitro and in vivo. Intein in tripartite fusion protein (EV-sorting Domain-Intein-Cargo) performs C-terminal cleavage during the process of EV-biogenesis, resulting in free cargo protein inside EVs. Together with a fusogenic protein, e.g. VSV-G, these engineered EVs achieve efficeint intracellular delivery of cargo protein (Cre and super repressor of NF-κB) or protein/RNA complex (Cas9/sgRNA RNPs) both in reporter cells and in mice models. **b** Fluorescence reporter construct expressed in the reporter cells generated to measure Cre delivery. **c** Constructs used for developing the VEDIC system. **d** Schematic of intein cleavage and intraluminal cargo release during EV biogenesis, MVB: multivesicular body. **e** Percentage of GFP positive reporter cells after incubating EVs for 2 days, as evaluated by flow cytometry. **f** Screen of fusogenic proteins co-expressed with CD63-Intein-Cre for EV-mediated delivery analyzed in T47D-TL cells after a two-day incubation period. **g**, Percentage of GFP positive reporter cells

after exposure to EVs derived from VSV-G co-transfected cells. **h** EV dose-and time-dependent recombination in HeLa-TL reporter cells mediated by VEDIC EVs. **i**, EV dose-and time-dependent recombination in B16F10-TL reporter cells mediated by VEDIC EVs. **j** Cre and VSV-G protein was analyzed in T47D-TL reporter cells by Western blot (WB) analysis, 48 hours (h) after addition of engineered EVs loaded with Cre in 24-well plates. **k** Comparison of the Cre transfer efficiency between VEDIC EVs and published Nanoblade system. Two-way ANOVA (Tukey) multiple comparisons test was used for analysis of (**g**) (**i**) and (**k**).; One-way ANOVA (Tukey) multiple comparisons test was used for analysis of (**f**). Experiments were done with 3 biological replicates except the Nanoblade (Cre) particles in (**k**) which had 2 biological replicates. Data are shown as mean ± SD. **a, d** and right panel of **k** Created in BioRender.com, with attribution line Zheng, W. (2025) https://BioRender.com/n1l657, Zheng, W. (2025) https://BioRender.com/i50r712 and Zheng, W. (2025) https://BioRender.com/n32h319 respectively. Exact statistical analysis was reported in the Source data and Source data are provided as a Source Data file.

function as an EV-sorting domain, similar to CD63. Hence, a VSV-G-Intein-Cre fusion protein was constructed, with and without a foldon component that has been demonstrated to enhance VSV-G trimer formation and function[30,31], and expression of constructs was confirmed (Supplementary Fig. 4d). Upon adding these EVs to reporter cells, a dose-dependent recombination in reporter cells was observed, and nearly 100% recombination in B16F10-TL and T47D-TL cells were detected in the high-dose groups (Fig. 2c, d and Supplementary Fig. 4a, b). The VSV-G-Foldon-Intein-Cre (VFIC) performed better than the VSV-G-Intein-Cre construct in low EV-dose experiments (Fig. 2c and Supplementary Fig. 4a, f), demonstrating the ability of the foldon domain to enhance VSV-G loading and/or function. Time-lapse fluorescence microscopy video analysis revealed similar recombination dynamics using VFIC EVs as was previously observed using the VEDIC system in HeLa-TL cells (Supplementary Movie #3). VFIC EV-mediated recombination was also observed in hard-to-transfect cells (Fig. 2e-h and Supplementary Fig. 4c). Morevoer, we purified VFIC EVs by ethier SEC or density gradient ultracentrifugation (DGUC) to see if their potency was maintained after purification. SEC-purified EVs showed comparable high-efficiency Cre delivery as the TFF-isolated EVs in the tested reporter cells in a dose-dependent manner (Fig. 2c, i, and Supplementary Fig. 4a, b). DGUC-purified VFIC EVs exhibited efficient time-dependent Cre transfer into HeLa-TL cells (Fig. 2j). We subsequently performed the direct co-culture, IBIDI, and Transwell assays using the VFIC system, and observed significant recombination in recipient cells by transfer of Cre protein from donor cells (Supplementary Fig. 3k and Supplementary Fig. 4e, g-i).

### The pH-sensitive intein performs C-terminal cleavage during EV-biogenesis

To confirm that the intein we used in this study performed C-terminal cleavage in a pH-dependent manner, we introduced 2 mutants (Fig. 3a). The H439Q variant has a lower pH-sensitivity, which should lead to a lower cleavage rate during EV-biogenesis, since the pH in MVBs is around 6 (Fig. 3a, middle panel)[32]. In contrast, the N440A variant is supposed to abolish the C-terminal cleavage which should lead to minimal cleavage both in EV-producing cells and the isolated EVs (Fig. 3a, lower panel)[33]. As expected, western blot showed a decreased cleavage of the mutants in line with the hypothesis (Fig. 3b). Accordingly, the functional assays (adding isolated EVs to recipient cells or direct co-culture with VEDIC producer cells and reporter cells) showed significantly decreased recombination in reporter cells when the 2 mutants were included in the VEDIC system (Fig. 3c-e, Supplementary Figs. 5a, b and 6a). These results were corroborated for the VFIC system (Supplementary Fig. 5c-g and Supplementary Fig. 6b, c). Altogether, these findings support the notion that the intein used for the VEDIC and VFIC technologies enables C-terminal cleavage to form soluble cargo proteins in the EV-lumen during EV-biogenesis in a pH-dependent manner (Fig. 1d).

### VSV-G boosts endosomal escape following receptor-mediated endocytosis in recipient cells

To ascertain the role of VSV-G in endosomal escape and endocytosis, we introduced mutations that have been described to disrupt fusogenic (P127D) or LDL-R receptor binding capacity (K47Q), respectively (Fig. 4a, b)[34–36]. In order to assess EV uptake and trafficking, Cre was replaced by mNG in the VEDIC system, which was co-transfected with VSV-G to produce EVs that were added to Huh7 cells[37]. Confocal microscopy was used to evaluate the uptake and delivery of the mNG cargo of the vesicles 48 h after incubation and showed a punctate distribution of fluorescent signal in recipient cells in the absence of VSV-G or when the P127D mutant was used, which co-localized with Rab5 and Rab7 positivie endosomes and thus indicating endosomal entrapment (Fig. 4c and Supplementary Fig. 7 and 8). However, when VSV-G was co-expressed, the mNG signal diffused into the cytosol (Fig. 4c), and co-localized with Rab5 and 7 poorly, suggesting endosomal escape. In line with our hypothesis, mNG was furthermore observed in punctate form and co-localized with Rab5 and 7 when CD63-mNG, without intein to liberate cargo protein, was co-transfected with VSV-G (Fig. 4d and Supplementary Fig. 7 and 8). In summary, functional VSV-G and liberated cargo was required for cargo release into the cytosol.

Next, the two mutant VSV-G constructs were used for Cre delivery to recipient cells and the Cre levels were analyzed by western blot. Decreased Cre protein levels were observed using western blot analysis for both mutants compared to the wide-type VSV-G group, when the same dose of EVs were added to the cells (Fig. 4e and Supplementary Fig. 9g)[36]. The western blot results were corroborated by recombination assays in the reporter cells with the two mutants in a VEDIC set up where the K47Q mutant showed a decreased activity and the P127D mutant a complete loss of reporter activity after EV treatment, direct co-culture, IBIDI and Transwell assays in different reporter cells (Supplementary Fig. 9a-f). Furthermore, we reproduced these results using the VFIC system (Fig. 4f-h and Supplementary Fig. 10).

### Robust gene editing by Cas9/sgRNA RNPs and meganuclease targeting PCSK9 using the VEDIC and VFIC systems

Since Cre and mNG proteins were successfully delivered, we next tested whether more relevant therapeutic cargoes could be delivered. To this end, Cre was replaced with Cas9 to encapsulate Cas9-ribonucleoprotein (Cas9-RNP) into engineered EVs (Fig. 5a, b and Supplementary Fig. 11b). As a read-out for Cas9-RNP delivery, we employed CRISPR reporter (Stoplight, SL) cells that constitutively express mCherry followed by a short linker region that includes a sgRNA target region and a stop codon, followed by two eGFP open reading frames (ORFs), +1nt and +2nt out of frame, respectively. Upon successful Cas9-RNP delivery, non-homology end joining (NHEJ)-mediate +1nt and +2nt frameshifts in the linker region will result in bypassing of the stop codon and the permanent expression of eGFP

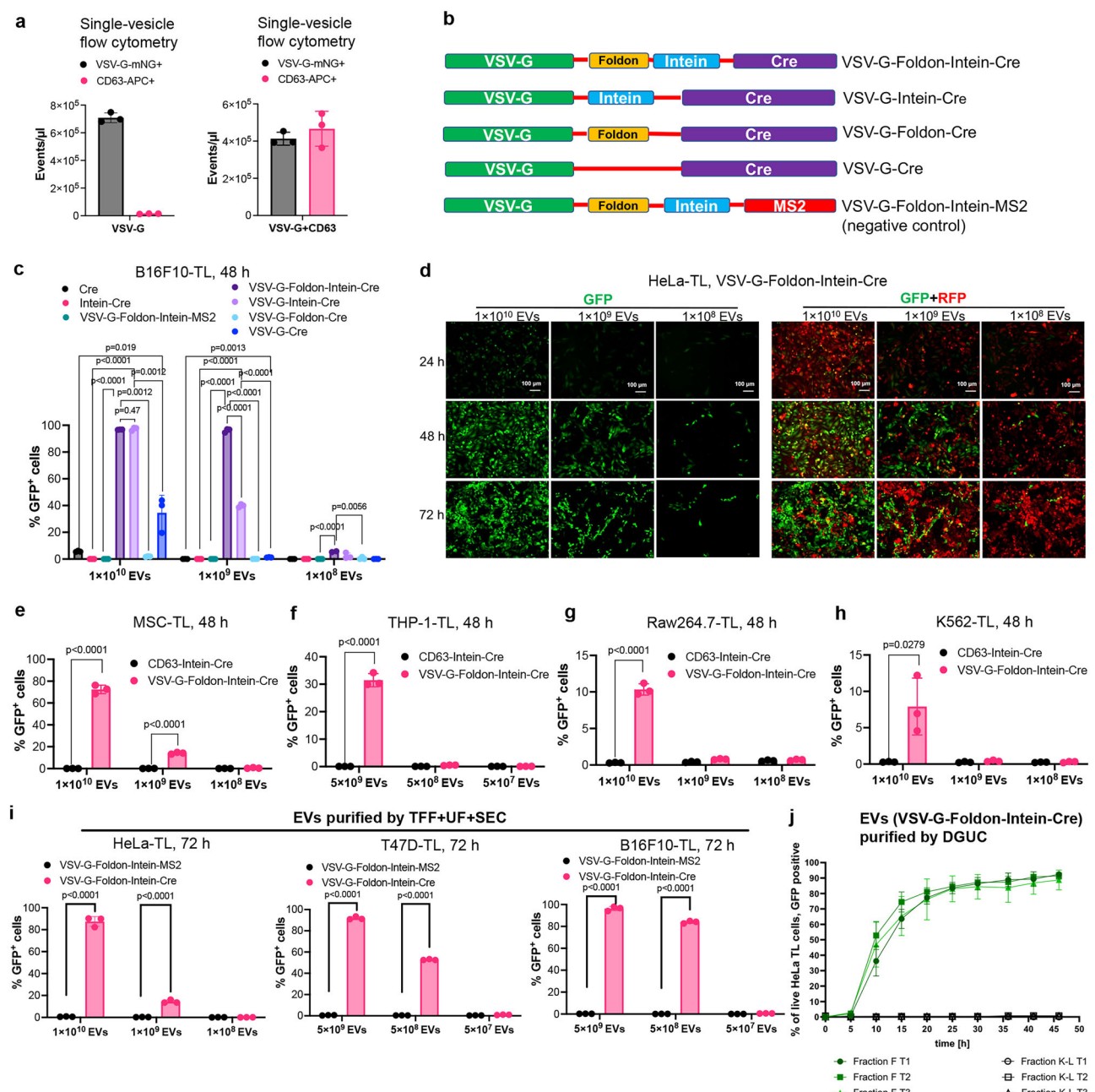

**Fig. 2 | Development of the VFIC system to further improve intracellular protein delivery by EVs. a** VSV-G+ and CD63 + EV concentrations as determined by single-vesicle flow cytometry after transfection with VSV-G-mNG alone or VSV-G-mNG and CD63 together. **b** Constructs generated for developing the VFIC system. The last construct was generated as a negative control where Cre was replaced with the bacteriophage protein MS2[49]. **c** EV dose-dependent recombination in B16F10-TL cells mediated by VSV-G-Foldon-Intein-Cre and VSV-G-Intein-Cre EVs as evaluated by flow cytometry. **d** Representative images showing GFP positive HeLa-TL cells 24, 48 and 72 h after exposure to VFIC EVs at different doses. Scale bar, 100 μm. **e−h** Recombination in hard-to-transfect reporter cells (MSC-TL, THP-1-TL,

Raw264.7-TL and K562-TL) mediated by VFIC EVs after 48 h. **i** The efficiency of Cre transfer in reporter cells by EVs isolated using TFF + UF + SEC method. TFF: tangential flow filtration; UF: ultrafiltration; SEC: size exclusion chromatography. **j** Dynamic Cre delivery in HeLa-TL cells using VFIC EVs purified by DGUC. DGUC: density gradient ultracentrifugation; T1-T3: technical replicates. Two-way ANOVA (Tukey) multiple comparisons test was used for analysis of (**c**), (**e−h**) and (**i**). Experiments were done with 3 biological replicates and data are shown as mean ± SD. Exact statistical analysis was reported in the Source data and Source data are provided as a Source Data file.

(Fig. 5c)[38]. Dose- and time-dependent genome editing was observed in cells treated with VEDIC and VFIC EVs (Fig. 5d, e and Supplementary Fig. 11a, d). For VFIC EVs, close to 80% gene editing efficiency was achieved, which is most likely the maximum achievable in the reporter cells (Fig. 5d)[38]. In addition, we directly compared the delivery efficiency of Cas9-RNP by VFIC EVs isolated by TFF only or further purified by SEC after TFF, and found no difference between these 2 methods (Supplementary Fig. 11e), which was consistent with the previous

results for Cre delivery (Fig. 2i). To confirm the dynamic change of proteins of the engineered EVs after endocytosis in HEK-SL cells, western blot was performed and as for Cre loaded EVs, VSV-G protein level increased with time and peaked 48 h after addition of EVs, and then decreased at 72 h (Supplementary Fig. 11c). Moreover, we quantified the Cas9 protein levels in the engineered VEDIC and VFIC EVs and found approximately 0.34 and 1.3 molecules per EV, respectively (Fig. 5f). To further corroborate the capability of efficient editing of the

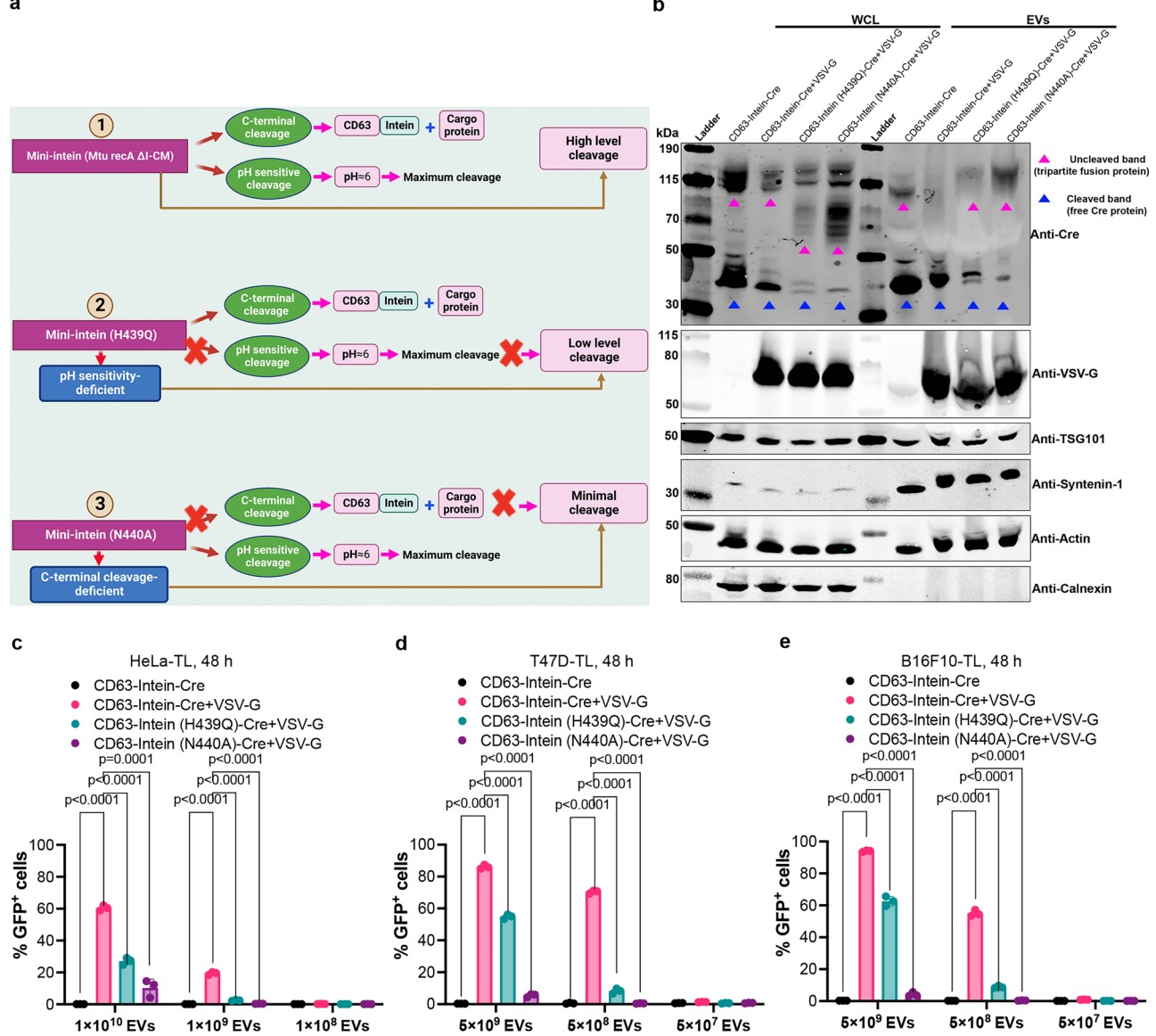

**Fig. 3 | The pH-sensitive intein performs C-terminal cleavage during EV-biogenesis. a** Schematic illustration of the cleavage mechanism of different engineered intein variants. **b** Protein expression of different engineered mutant intein constructs in whole cell lysates (WCL) and isolated EVs derived from HEK293T cells evaluated by western blot analysis. Lysates from $5 \times 10^5$ EV-producing cells and $1 \times 10^{10}$ engineered EVs were loaded onto the western blot gel. TSG101, syntenin-1 and β-actin were used as EV markers and Calnexin was used as a cellular organelle marker (endoplasmic reticulum) that should be absent in EV samples.

**c–e** Recombination in reporter cells mediated by EVs derived from engineered cells using mutant inteins (H439Q and N440A) in VEDIC system. Two-way ANOVA (Tukey) multiple comparisons test was used for analysis of (**c–e**). Experiments were done with 3 biological replicates and data are shown as mean ± SD. **a** Created in BioRender.com, Zheng, W. (2025) https://BioRender.com/v41e325. Exact statistical analysis was reported in the Source data and Source data are provided as a Source Data file.

VFIC system for Cas9-RNP delivery, the endogenous gene, mouse transthyretin (mTTR) was targeted. N2a cells incubated with engineered EVs showed 80% indels analyzed by Sanger sequencing and ICE analysis (Fig. 5g). Next, VEDIC and VFIC constructs for the delivery of a previously described meganuclease targeting *PCSK9* were generated (Fig. 5h)[39]. Upon EV exposure to cells, a significant decrease of PCSK9 protein level was observed in a dose-dependent manner (Fig. 5i).

### Cre-mediated recombination in melanoma-xenograft and R26-LSL-tdTomato reporter mice by VEDIC and VFIC systems after local injection

Based on the promising results achieved in vitro, the in vivo applicability of VEDIC and VFIC-engineered EVs was assessed. To evaluate VEDIC and VFIC-mediated Cre delivery in vivo, an intratumoral (IT)

injection of engineered EVs was conducted in C57BL/6 mice bearing subcutaneous B16F10-TL melanoma xenografts (Supplementary Fig. 12a). 4 days after injection, tumors were harvested for immunohistochemistry analysis, which showed significant GFP signals following treatment with VEDIC and VFIC EVs (Supplementary Fig. 12b). Next, R26-LSL-tdTomato reporter mice, whereby successful Cre delivery excises a stop cassette that are present between the CAG promoter and an ORF of the red fluorescent protein, tdTomato, were exploited for in vivo studies. These mice were injected with VEDIC and VFIC EVs through intracerebroventricular (ICV) injection as a single dose and one week later the Cre delivery in the brain was assessed (Fig. 6a)[26]. As shown in Fig. 6b, tdTomato expression was detected in the cerebellum, cortex, and hippocampus of VEDIC- and VFIC EV treated mice. Moreover, we additionally observed trace amounts of tdTomato expression

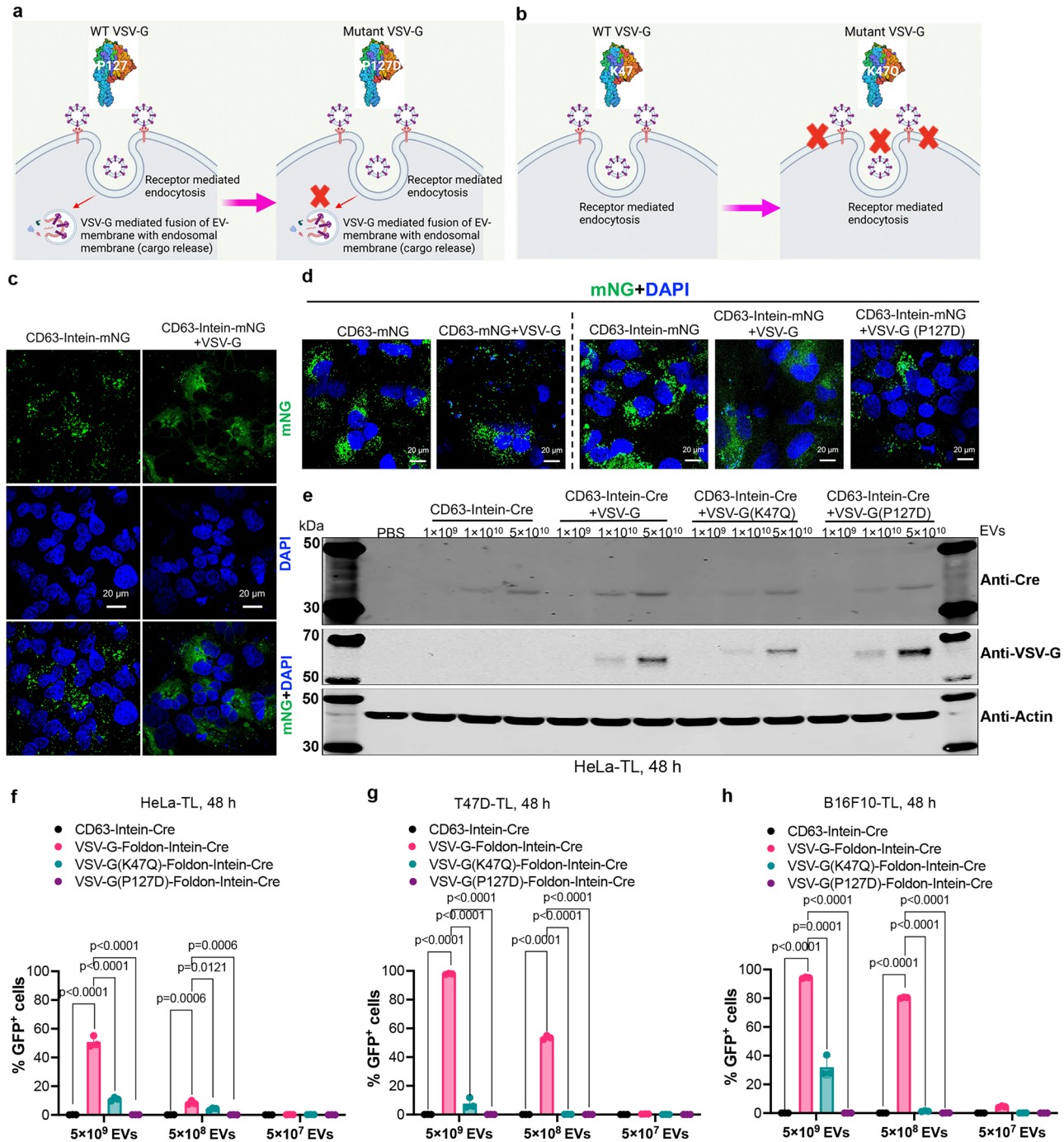

**Fig. 4 | VSV-G boosts endosomal escape following receptor-mediated endocytosis of engineered EVs into recipient cells. a, b** Properties of the two VSV-G mutants: VSV-G P127D loses the capacity to mediate fusion between the EV-and endosomal membranes and VSV-G K47Q has a reduced capacity to bind to LDLR on the cell surface. **c** Confocal immunofluorescence demonstrating the subcellular distribution of mNG in the presence or absence of wild type VSV-G engineered EVs in Huh7 cells. Scale bar, 20 μm, representative images. **d** Subcellular distribution of mNG in different groups after adding the indicated engineered EVs determined by confocal immunofluorescence. Scale bar, 20 μm, representative images. **e** WB evaluation of protein levels of Cre and VSV-G in HeLa-TL reporter cells after addition of engineered EVs with wild type, P127D or K47Q VSV-G in 24-well plates. (**f–h**) Percentage of GFP positive HeLa-TL, T47D-TL and B16F10-TL cells after adding wild type, P127D or K47Q VSV-G plus CD63-Intein-Cre EVs, as evaluated by flow cytometry. Two-way ANOVA (Tukey) multiple comparisons test was used for analysis of (**f–h**). Experiments were done with 3 biological replicates and data are shown as mean ± SD. **a, b** Created in BioRender.com, Zheng, W. (2025) https://BioRender.com/n76v160 and Zheng, W. (2025) https://BioRender.com/n89p785 respectively. Exact statistical analysis was reported in the Source data and Source data are provided as a Source Data file.

in the olfactory bulb and thalamus of VEDIC and VFIC treated animals (Supplementary Fig. 13a). To further identify which cell types internalized the engineered EVs, the slides were co-stained for both tdTomato and specific cell-marker genes. We found good colocalization of tdTomato with GFAP (astrocyte marker) and IBA1 (microglia marker), but only marginal colocalization of NeuN (neuronal marker) in the corpus callosum and hippocampus in the engineered EV-treated animals (Supplementary Fig. 13b-f). Based on the above results, we concluded that the engineered EVs mainly delivered their cargo to astrocytes and microglia in the brain.

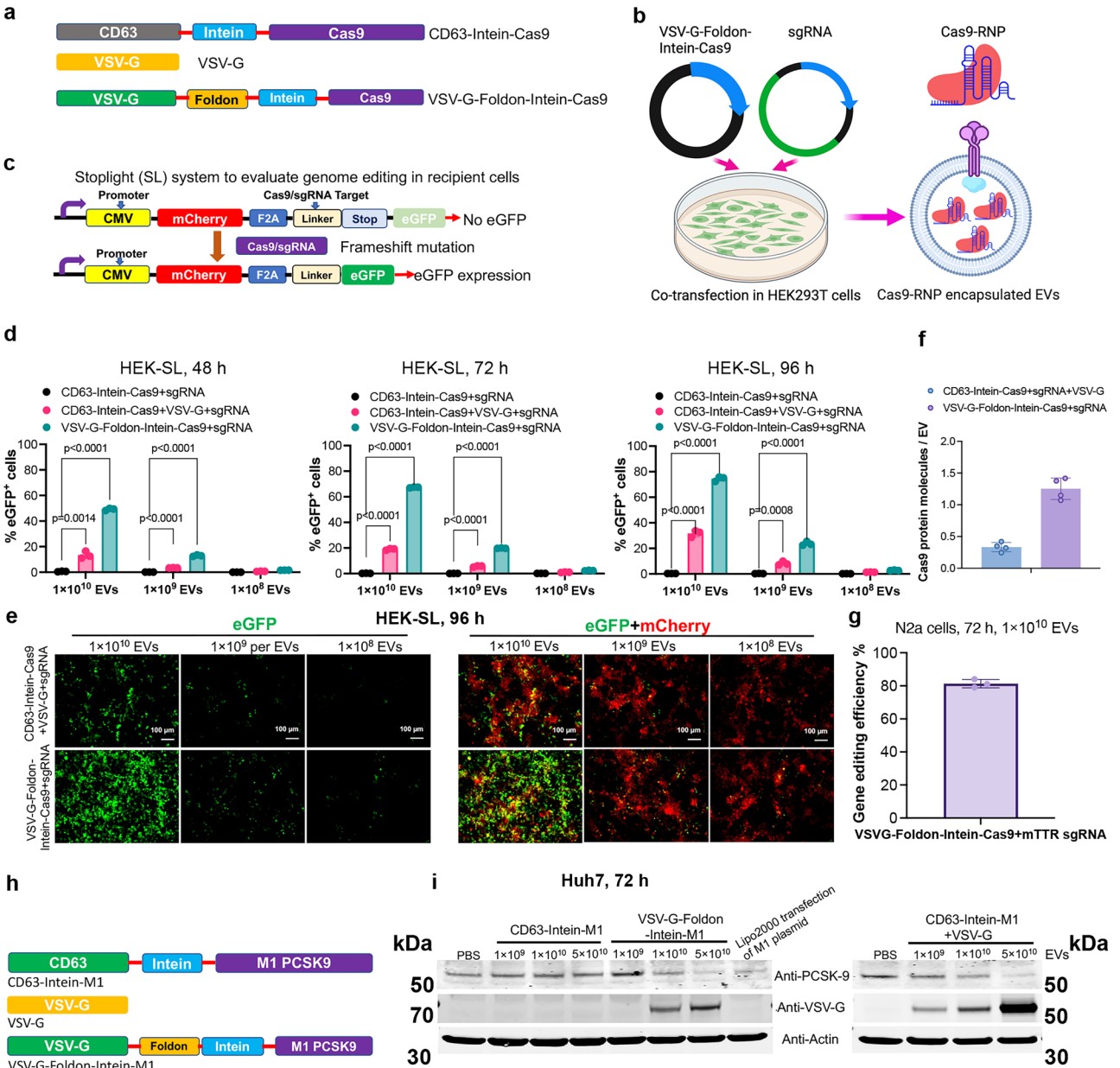

**Fig. 5 | Robust gene editing by Cas9/sgRNA RNP and meganuclease targeting PCSK9 using the VFIC and VEDIC systems. a** Constructs generated for Cas9/sgRNA RNP delivery. **b** Schematic illustration regarding Cas9/sgRNA RNP encapsulation into engineered EVs. **c** Schematic of reporter cells used to functionally assess Cas9/sgRNA RNP delivery by engineered EVs. **d** Percentage of eGFP positive cells after addition of engineered EVs, as measured by flow cytometry 48, 72 and 96 h after EV addition. **e** Immunofluorescence demonstrated gene-editing in recipient cells after treatment with different doses of engineered EVs after 4 days. Scale bar, 100 μm, representative images. **f** Cas9 protein quantification in the engineered

VEDIC and VFIC EVs. **g** Editing efficiency for the endogenous target mTTR in N2a cells by engineered EVs evaluated by Sanger sequencing. **h** Constructs generated for EV-mediated delivery of meganuclease targeting PCSK9. **i** WB analysis of PCSK9 and VSV-G protein in Huh7 cells after treatment with different doses of EVs in 24-well plates. Two-way ANOVA (Tukey) multiple comparisons test was used for analysis of (**d**). Experiments were done with 3 biological replicates and data are shown as mean ± SD. **b** Created in BioRender.com, Zheng, W. (2025) https://BioRender.com/g77t273. Exact statistical analysis was reported in the Source data and Source data are provided as a Source Data file.

To enhance functional CNS delivery, we next repeated the ICV experiments using osmotic minipumps, allowing for a stable infusion of engineered EVs over a 24 h time period (Fig. 6c). Strikingly, upon flow cytometry analysis of single cells 4 days after the infusion, up to 40% and 30% of cells in hippocampus and cortex respectively were edited by Cre delivered via the VEDIC system (Fig. 6d and Supplementary Fig. 14a). In addition, close to 10% cells in cerebellum were also edited by VEDIC EVs (Fig. 6d). In order to rule out any potential neuroinflammatory response from the EV infusion, we performed H&E (hematoxylin-eosin) staining and IHC staining of p65 in different brain regions (hippocampus, cortex and cerebellum), and found no obvious

inflammatory cell infiltration and p65 overexpression after osmotic ICV minipump injection of EVs (Supplementary Fig. 14b, c).

### Cre recombination in R26-LSL-tdTomato reporter mice following systemic VEDIC and VFIC EV-mediated Cre delivery

Next, one pilot ex vivo study was performed by adding engineered EVs to liver primary cells harvested from R26-LSL-tdTomato reporter mice, which showed significant recombination of the cells by VEDIC and VFIC EVs (Supplementary Fig. 15a). Next, engineered EVs were administered via intraperitoneal (IP) injection into R26-LSL-tdTomato reporter mice. The liver, spleen and heart were harvested for analysis by

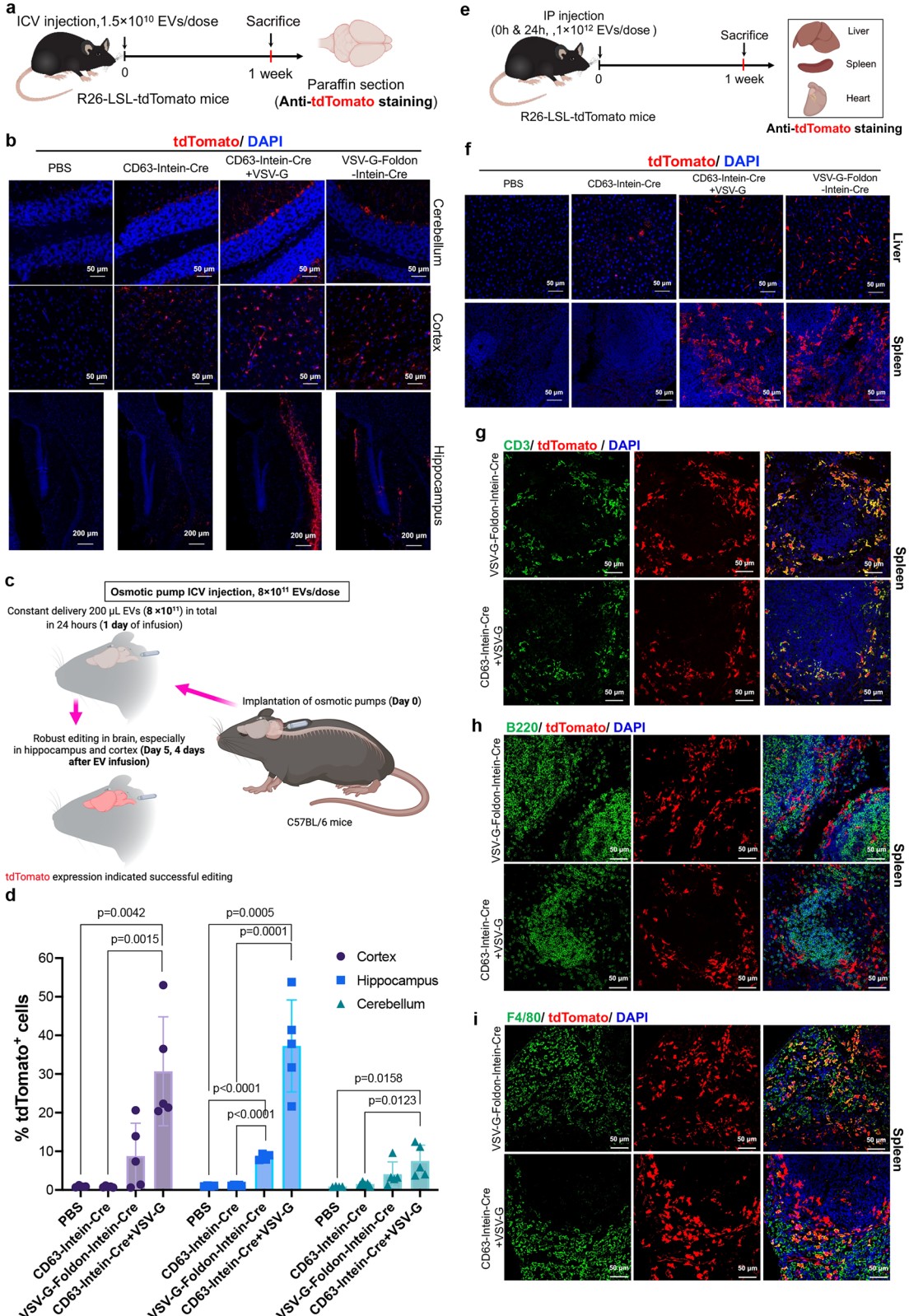

immunofluorescence one week after injection (Fig. 6e). A substantial number of cells in the liver and spleen were found to be tdTomato positive following injection of both VEDIC and VFIC EVs, but not after injection of CD63-Intein-Cre EVs (Fig. 6f). In contrast, we did not observe any significant tdTomato expression in heart, known to be hard to reach with HEK293T derived EVs (Supplementary Fig. 15b).

Co-staining of tdTomato and cell specific markers in spleen revealed a high degree of functional delivery to leukocytes (CD45 + ), especially to T cells (CD3 + ) and macrophages (F4/80 + ), was detected (Fig. 6g, i and Supplementary Fig. 15c, d, e, g). In contrast, B cells (B220 + ) had a low recombination efficency (Fig. 6h and Supplementary Fig. 15f).

**Fig. 6 | Cre recombination in R26-LSL-tdTomato reporter mice by VEDIC and VFIC engineered EVs after local (ICV or osmotic pump ICV) and systemic (IP) injections. a** Workflow for the intracerebroventricular (ICV) injection of engineered EVs to deliver Cre in CNS of R26-LSL-tdTomato reporter mice. **b** TdTomato expression analyzed by immunofluorescence in different regions of CNS after ICV injection of engineered EVs. Scale bar, 50 μm for cerebellum and cortex, and 200 μm for hippocampus. **c** Workflow for the osmotic pump ICV injection of engineered EVs to transfer Cre to CNS in R26-LSL-tdTomato reporter mice. **d** Percentage of tdTomato+ cells in the brain tissues after osmotic pump ICV injection of engineered EVs quantified by flow cytometry, analyzed 4 days after the infusion. *n* = 4 mice for PBS group and *n* = 5 mice for other groups. **e** Schematic workflow for the intraperitoneal (IP) injection of engineered EVs into R26-LSL- tdTomato reporter mice. **f** TdTomato expression in liver and spleen after IP injection of engineered EVs, analyzed one-week post injection. Scale bar, 50 μm. **g** Co-staining of tdTomato and the T cell marker CD3 in spleen as detected by immunofluorescence one week after IP injection of engineered EVs. Scale bar, 50 μm. **h** Co-staining of tdTomato and the B cell marker B220 in spleen one week after IP injection of engineered EVs. Scale bar, 50 μm. **i** Co-staining of tdTomato and the macrophage marker F4/80 in spleen one week after IP injection of engineered EVs. Scale bar, 50 μm. *n* = 3 mice per group, representative images for (**b**) and (**f**–**i**). Two-way ANOVA (Tukey) multiple comparisons test was used for analysis of (**d**). Data are shown as mean ± SD. **c** Created in BioRender.com, Zheng, W. (2025) https://BioRender.com/z93x284. Exact statistical analysis was reported in the Source data and Source data are provided as a Source Data file.

## Treatment of Lipopolysacharide (LPS)-induced systemic inflammation by VEDIC and VFIC-mediated delivery of super-repressor of NF-κB

To demonstrate the applicability of our systems for the treatment of disease, we applied the VEDIC and VFIC EVs to treat lipopolysaccharide (LPS)-induced systemic inflammation by delivering a previously reported super-repressor of NF-κB activity (SR) (Supplementary Fig. 16a)[40]. To accomplish this, CD63-Intein-SR, VSV-G-Intein-SR and VSV-G-Foldon-Intein-SR constructs were generated, and HEK-NF-κB luciferase reporter cells were used as a read-out for functional in vitro assessment of the system (Fig. 7a, b and Supplementary Fig. 16d). The reporter cell is equipped with a luciferase reporter that is controlled by a minimal NF-κB promoter; hence it is activated by stimuli that activates the NF-κB pathway such as LPS or TNF-α. The super-repressor of NF-κB activity (SR) will block the translocation of NF-κB into the nucleus and thus block NF-κB signaling and the cells can thus be used as a screening tool for SR delivery (Fig. 7c)[41]. First, we confirmed that the NF-κB SR, which is constitutively active and maintains the NF-κB in the cytoplasm, inhibited the HEK-NF-κB luciferase reporter activation (Supplementary Fig. 16b, c)[42]. SR delivered by VSV-G-Foldon-Intein-SR, VSV-G-Intein-SR, and VSV-G + CD63-Intein-SR EVs successfully inhibited TNF-α-mediated NF-κB signaling activation (Fig. 7d, e). This provided the rationale for testing these EVs in a murine LPS-induced model of systemic inflammation (Fig. 7f). Thus, engineered EVs were injected 4 h before and 6 h after injecting LPS to allow for, and enhance, the binding of EV-delivered SR to NF-κB, respectively. Subsequently, the weight and mortality of the mice were measured at 24 and 48 h after LPS treatment. Compared to the PBS and CD63-Intein-SR injected groups, the body weight and survival of VSV-G + CD63-Intein-SR and VSV-G-Foldon-Intein-SR treated animals was significantly improved after 48 h (Fig. 7g, h). Histology of the liver revealed a decrease of inflammatory cells at the portal areas as well as a significant alleviation of the hydropic degeneration of hepatocytes by treatment with both VEDIC and VFIC engineered EVs (Fig. 7i). Concurrently, a decrease in leukocyte infiltration in the para glomerulus area was found in kidneys treated with both VEDIC and VFIC engineered EVs (Fig. 7j). Furhtermore, protective role of both VEDIC and VFIC engineered EVs was also identified in lungs as demonstrated by less inflammatory cells infiltration in mice lung tissues of these 2 groups (Supplementary Fig. 16e). These results demonstrate the adaptability and therapeutic potential of the VEDIC and VFIC systems for the treatment of disease.

## Discussion

By fine-tuning our engineering strategies, we have identified solutions to the major bottlenecks of EV mediated delivery or protein therapeutics, the enrichment of liberated active cargo into EVs and their subsequent endosomal escape in recipient cells. The resulting VEDIC and VFIC systems achieved an unprecedented level of efficiency for EV-mediated intracellular delivery of functional proteins in vitro and in vivo. Owing to the versatility of the VEDIC and VFIC systems, we anticipate that many other therapeutic proteins or protein complexes of interest can be efficiently delivered, apart from Cre, super-repressor of NF-κB and Cas9-RNPs explored in this work. As such, these approaches may hold great potential for further therapeutic development.

Importantly, up to 40% and 30% cells in hippocampus and cortex respectively were edited by Cre delivered via VEDIC system after osmotic pump ICV infusions during 24 h, the efficiency of which is even comparable to the traditional AAV-mediated DNA delivery[43,44]. Encouraged by this, we anticipate the development of therapeutics for central nervous system (CNS) genetic diseases, such as Huntington's disease and spinal muscular atrophy[45,46], through delivery of gene editing tools (CRISPR/Cas9 or base editors) to the CNS using engineered EVs.

Furthermore, we applied engineered EVs for the treatment of LPS-induced systemic inflammation, demonstrating that therapeutic levels of intracellular protein delivery were achieved by our systems in vivo. These observations demonstrate the therapeutic potential of these approaches, which shows exciting promise for the potential development of treatments for a wide array of pathologies, such as lysosomal storage diseases (LSDs) and enzymatic deficiencies[47,48]. To summarize, the VEDIC and VFIC systems developed in this study allow for robust intracellular functional delivery of therapeutic proteins, both in vitro and in vivo. In addition, the high genome editing efficiency achieved by Cas9-RNP delivery implies that this strategy may lead to potential applications in the treatment of genetic diseases.

## Methods

### Ethical statement

All mouse experiments were performed in accordance with the ethical permission granted by The Swedish Local Board for Laboratory Animals and designed to minimize the suffering and pain of the animals, with ethical permit number (LPS model:16212-2020; tumor model:2173-2021); or approved by Inflammation Research Center, Ghent University with ethical permit number: LA1400091/LA2400526.

### Cell lines

HEK293T cells used to produce the functional EVs in this study were maintained in Dulbecco's Modified Eagle Medium (DMEM) (high glucose) supplemented with 10% fetal bovine serum (FBS) (Gibco, USA) and 1% Antibiotic-Antimycotic (Anti-anti) (Gibco, USA). Cells were cultured at 37 °C in a humidified air atmosphere containing 5% CO₂. The reporter cell lines (Hela-TL, T47D-TL, B16F10-TL, Raw264.7-TL, HEK-NF-κB and HEK293T-SL) were cultured using the same medium and the same conditions as HEK293T cells. THP-1-TL, K562-TL and MSC-TL cells were cultivated in Roswell Park Memorial Institute (RPMI) 1640 medium (high glucose) supplemented with 10% FBS and 1% Anti-anti. The information about cell lines was provided in Supplementary Table 6.

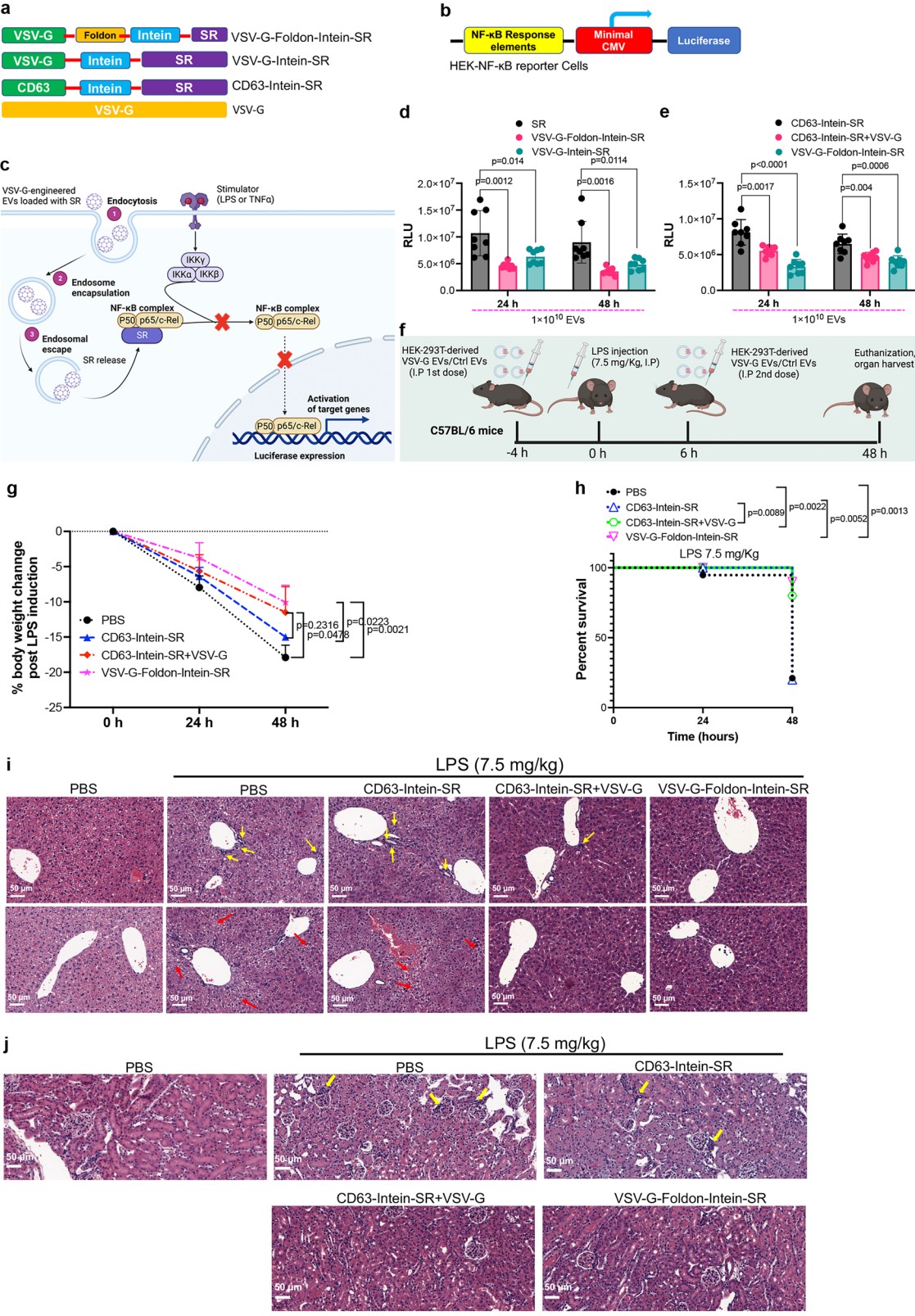

## Mice model (Intra-tumor injection)

C57BL/6 mice (substrain C57BL/6 J, 5 weeks of age, 20 g body weight, n = 3 mice per group, in total 12 mice, all are female) were acclimated to their new surroundings (ambient temperature: 20–22 °C, humidity: 45–55%, dark/light cycle: 12/12 h) at least one week before the experiment. B16F10-TL cells resuspended in PBS were inoculated subcutaneously into the mice at a density of 0.5 million cells per mouse. Ten days after inoculation when obvious tumors were formed, engineered EVs were injected directly into the tumors. The injected volume was 50 μL per mouse with $7.5 \times 10^{10}$ EVs. Four days after intra-tumor injection of EVs, the mice were sacrificed, and tumors were harvested and fixed in PFA. Tumor tissues were stained

**Fig. 7 | Treatment of LPS-induced systemic inflammation using VEDIC and VFIC-EV mediated delivery of a super-repressor of NF-κB. a, b** Design of the constructs and reporter cells utilized for delivery and assessment of a super-repressor of NF-κB by engineered EVs. **c** Schematic illustration how the EV-delivered super-repressor of NF-κB inhibits NF-κB activity. **d, e** Luciferase activity from HEK-NF-κB reporter cells, 24 or 48 h after treatment with engineered EVs (TNF-α stimulation for 6 h before harvesting cells), in 24-well plates. **f** Schematic illustration of the workflow for the treatment of LPS-induced systemic inflammation by engineered EVs in mice ($5 \times 10^{11}$ EVs/mouse per dose). **g, h** Percentage of body weight loss and group survival in mice after LPS and engineered EV injections. $n = 10$ mice per group. **i** Representative histology images (hematoxylin-eosin stain) of liver to show the aggregation of inflammatory cells (upper panel, yellow arrows indicate the aggregated inflammatory cells in portal areas) and the hydropic degeneration of hepatocytes (lower panel, red arrows indicate the hydropic degeneration of hepatocytes) after LPS induction. Scale bar, 50 μm. **j** Accumulation of inflammatory cells in the proximal region of renal tubules as shown by the representative histology images (yellow arrows indicate the inflammatory cells). Scale bar, 50 μm. Two-way (Tukey) ANOVA multiple comparisons test was used for analysis of (**d, e**, and **g**); Log-rank (Mantel-Cox) test was used for the analysis of survival curve (**h**). Experiments were done with 8 biological replicates for (**d** and **e**) and data are shown as mean ± SD. **c, f** Created in BioRender.com, Zheng, W. (2025) https://BioRender.com/s30x956 and Zheng, W. (2025) https://BioRender.com/j46d117 respectively. Exact statistical analysis was reported in the Source data and Source data are provided as a Source Data file.

immunohistochemically for GFP expression while tissues in lysis buffer were homogenised using a tissue lyser machine. Animal information was provided in Supplementary Table 7.

### Mice model (LPS-induced inflammation)

C57BL/6 mice (substrain C57BL/6 J, 5 weeks of age, 20 g body weight, $n = 10$ mice per group, in total 40 mice, all are female) were acclimated to their new surroundings (ambient temperature: 20–22 °C, humidity: 45–55%, dark/light cycle: 12/12 h) at least one week before the experiment. Animals were injected (IP) with engineered EVs 4 h before IP injection of LPS (Sigma, USA) at the dose of 7.5 mg/Kg. Six hours after LPS induction, engineered EVs were IP injected once more to boost the intracellular delivery of the protein cargos by EVs. The survival rate and body weight of the mice with LPS induction were recorded for 2 days. 48 hours after LPS induction, the mice were euthanized and sacrificed, and main organs, such as liver, were harvested and fixed with PFA. H&E (haematoxylin and eosin) staining was performed to check the extent of damage to the organs induced by LPS. The damage of the tissues was evaluated by a professional pathologist. Animal information was provided in Supplementary Table 7.

### Mice model (ICV delivery of EVs via osmotic pump)

For a constant intracerebroventricular delivery of the EV preparations, Alzet® osmotic pumps (2001D) were used. These pumps were filled with 200 μl of the EV preparation at a concentration of $4 \times 10^{12}$ EVs/ml. The osmotic pumps were prepared and implanted as described by Sanchez-Mendoza and colleagues. The mice were kept at 20–22 °C, with 40–60% humidity and 14 hours of light/10 hours of night cycle. In short, 15-17 weeks old R26-LSL-tdTomato reporter mice ($n = 4$ for PBS group and n = 5 for other 3 groups, in total 19 mice, female = 16 and male = 3, based on C57BL/6 J and substrain is B6.Cg-Gt(ROSA) 26Sor*tm9(CAG-tdTomato)Hze*/J (*Rosa26.tdTomato*)) reporter mice were anesthetized with isoflurane and mounted on a stereotactic frame (the same type of mice were used for ICV and IP injections). A constant body temperature of 37 °C was maintained using a heating pad. Next, a small incision in the skin was made from the base of the neck to up in between the eyes. The injection needle of the pump system was placed in the ventricles based on coordinates measured relative to the bregma intersection (anteroposterior 0.07 cm, mediolateral 0.1 cm, dorso-ventral 0.2 cm). The cannula was fixed to the skull with Loctite 454. A subcutaneous pocket for the osmotic pump was made by using a blunted scissor to slide underneath the skin at the base of the neck. Next the pumps were inserted under the skin at the base of the neck and pushed to the back as far as was possible without resistance. The osmotic pump was connected to the cannula via a vinyl catheter (0.71 mm outer diameter). An immediate constant delivery of the EV preparations started upon implantation of the pumps at a flow rate of 8 μl/hour. After surgery the incision was sutured and 48 hours after the implantation, the pumps were removed, and bone wax was used to close the skull again. Animal information was provided in Supplementary Table 7.

### Flow cytometry of single cells from brain tissues

The terminal sedation of mice was performed with I.P administration of Ket/Xyl (Ket/Xyl mix=100 mg/kg ketamine; 20 mg/kg xylazine). And then blood was collected from the right ventricle, and PBS/heparin was used for trans cardinal perfusion (10 ml per mouse at rate of 3 ml/min). Next, the brain samples were collected in 400 μl serum free RPMI medium at 4 °C, followed by adding 200-400 μl enzyme mix (Enzyme mix = 10 U/mL collagenase I, 400 U/mL collagenase IV, and 30 U/mL DNaseI (Worthington) in PBS). In the next step, the brain samples were cut into small pieces with scissor, and incubated for 10 min at 37 °C in oven without shaking. Pipette up and down the samples and incubate 10 min at 37 °C in oven without shaking, and repeat this for another 2 times. And then put the single cell mixture on a cell strainer (70 μm, Falcon) and collect cells into a new Eppendorf 1.5 ml tubes for pellet. Rinse the tubes with 500 μl RPMI medium and put this also onto the cell strainer, and spin down the cells at 400 g for 7 min at 4 °C. Resuspend the cells in 40% percoll in HBSS (2 ml solution=1.2 ml HBSS + 0.8 ml percoll). And then spin down the cells at 600 g for 10 min at 4 °C without breaks. Resuspend the cell pellet in staining buffer and spin down the cells again at 600 g for 10 min at 4 °C. At last, resuspend the cell pellet in PBS and let the cells go through the cell strainer again before flow cytometry. Run 500,000 cells for each sample on the flow and the data was analyzed by using FlowJo.

### Construct generation

All the transgenes, except the ones purchased from Addgene and the human-derived fusogens, were ordered from IDT (Integrated DNA Technologies, USA). The transgenes were first cloned into the pLEX vector backbone using EcoRI and XhoI sites. The constructs used in this study were then generated from the ordered fragments through restriction enzyme digestion and subsequent self-ligation. VSV-G-Intein-Cre and VSVG-Foldon-Cre were generated from the VSV-G-Foldon-Intein-Cre construct using Kpn2I and MluI, respectively. For VSV-G-Cre, VSV-G-Intein-Cre was digested with MluI. CD63-Intein-RS was generated by digesting VSV-G-Foldon-Intein-RS and CD63-Intein-Cre with BamHI and XhoI, and then inserting RS into the resulting CD63-Intein vector. Similarly, to synthesize CD63-Intein-Cas9 and VSV-G-Foldon-Intein-Cas9 constructs, CD63-Intein-Cre and VSV-G-Foldon-Intein-Cre were digested with BamHI and XhoI, and Cas9 inserted into CD63-Intein and VSV-G-Foldon-Intein, respectively. The CD63-Intein-M1 and VSV-G-Foldon-Intein-M1 constructs were generated by replacing Cre with M1 (M1 PCSK9 meganuclease) using BamHI and XhoI digestion. The 40 human-derived fusogens were ordered from Twist (Twist Bioscience) and the vector used was pTwist CMV BetaGlobin. Detailed information about the plasmids used are provided in Supplementary Table 3 and Supplementary Table 4.

### Plasmid transfection

HEK293T cells were seeded into 15-cm dishes, with the numbers of dishes decided according to the amount of EVs to be used in indicated experiments. Polyethylenimine (PEI, Polysciences) was utilized for the

transfection of plasmid/s according to the protocol provided by the manufacturer. The ratio of PEI to plasmid was 2:1 in this study. For single plasmid transfection, 30 μg plasmid was used for each plate. For co-transfection of 2 plasmids, 20 μg of each plasmid was used while for 3 plasmids, 15 μg of each plasmid was used.

## EV production

EVs were produced by transient transfection of the transgenes using polyethylenimine. HEK293T cells were seeded into 15-cm dishes at a density of 5 million cells per dish using complete DMEM medium. After 2 days, the cells were transfected with the transgenes and the medium changed to Opti-MEM (Gibco, USA) with 1% Anti-anti 6 h post-transfection. After 48 h, the conditioned medium (CM) was collected and centrifuged (700x $g$ for 5 min followed by 2000x $g$ for 10 min). The supernatant was then filtered through a 0.22 μm filter system.

## VLPs (Nanoblade) production

The plasmid pBIC-Gag-CAS9 was purchased from Addgene (#119942) and pBIC-Gag-Cre was generated by In-Fusion cloning, replacing CAS9 with Cre recombinase. The other plasmid pBS-CMV-gagpol for VLP production was also ordered from Addgene (#35614). BaEVRless was ordered as a fragment after codon optimization and cloned into pLEX vector by using EcoRI and XhoI restriction sites. HEK-293T cells were seeded into 15-cm dishes and then were transfected by plasmids (one dish: 3.4 μg pBIC-Gag-Cre; 5.6 μg pBS-CMV-gagpol; 1.4 μg BaEVRless and 0.8 μg VSV-G) 2 days after by using PEI. The medium was changed to Opti-MEM (Gibco, USA) with 1% Anti-anti 6 h after transfection and condition medium was collected after 48 h transfection. And then the medium was centrifuged at 700 g for 5 min and subsequent 2000 g for 10 min. After centrifugation, the medium was filtered with 0.45 μm filtration system and then the filtered medium was run TFF to isolate the EVs. After TFF, the isolated EVs were concentrated by using ultra-filtration and concentrations were determined by using Zetaview (Particle Metrix, Germany).

## EV isolation

Tangential flow filtration (TFF, MicroKross, 20 cm², Spectrum labs) was used to isolate EVs from the filtered CM. Particles greater than the 300 kDa cutoff of the TFF were retained in the system and concentrated. These particles were further concentrated using Amicon Ultra-15 100 kDa (Millipore) spin filters, which were centrifuged at 4000x $g$ for 30 min to several hours at 4 °C, depending on the amount EVs in the samples. Lastly, the concentrated EVs were collected in maxirecovery 1.5 ml Eppendorf tubes (Axygen, USA) and quantified using Nanoparticle Tracking Analysis (NTA).

## Size Exclusion Chromatography

500 μl concentrated EV samples after TFF were used for further purification by size exclusion chromatography. Briefly, the 70 nm, 500 μl qEV columns (IZON Science LTD) were primed with PBS and then followed by adding 500 μl EV samples to go through and collecting the solution with 15 ml falcon tubes. Discard the first 3 ml solutions which were supposed to be PBS, and collect the subsequent 2.5 ml samples which were supposed to be the EV fractions. Wash the columns with enough PBS and get rid of the potential residual contamination by using 0.5 M NaOH to go through the columns. And at last wash out the residual NaOH and keep the columns in the fridge. The 2.5 ml EV samples were further concentrated by using Amicon Ultra-2 10 kDa (Millipore) spin filters and the concentrations were determined by using Zetaview (Particle Metrix, Germany).

## Density gradient ultracentrifugation (DGUC)

DGUC gradient for VSV-G-Foldon-Intein-Cre EVs was prepared as follows: PBS 1 ml, 18% OptiPrep 1.5 ml, 23% OptiPrep 0.5 ml, 30% OptiPrep 3 ml, 45% OptiPrep 6 ml into 12 ml ultracentrifuge tube (TH-641 rotor

for 16 h at 29 600 rpm at 4 °C with acceleration = 4 and deceleration = 4). The next day gradient fractions were harvested by 500 μl each and labelled A- M (first 6.5 ml). Fraction "F" is specifically the EV fraction and fraction "K-L" refers to the aggregates.

## EV functionality assay by DGUC-derived EVs

HeLa-TL cells were seeded in DMEM + 10% FBS at 4000/well onto 384 well plate (PhenoPlate, Perkin Elmer), one day before the treatment. On the day of the experiment, cells were stained with Hoechst (10 min at 37 °C), and washed twice with Opti-MEM. Treatments were prepared to a final concentration of $9 \times 10^{10}$ particles/ml. Working stocks were prepared in triplicates (3 technical replicates), providing for treatment of 3 wells with 100 μl from each (3 biological replicates within). Media was aspirated from the PhenoPlate. And then the treated plate was inserted into Operetta HCS instrument and time-lapse images were acquired in 10 timepoints every 5 hours. Acquired channels: "eGFP" 200 ms at 80% power, "Hoechst 33342" 100 ms at 100% power, "RFP" 200 ms at 80% power, binning =1. For the analysis – max projection of 6 adjacent z-planes (0.5 um apart), was analysed in Harmony software 4.9 (Perkin Elmer): blocks used "find nuclei","find cytoplasm", and cells expressing eGFP following Cre activity were selected and counted when having mean eGFP fluorescence intensity of a cell >220 ("green cells"), red cells were counted when mean eGFP fluorescence intensity of a cell <=220 and RFP mean fluorescence intensity of a cell > 220; total living (expressing) cells were counted as the sum of "green" and "red" cells. Graphs were potted in GraphPad Prism.

## Nanoparticle Tracking Analysis (NTA)

EV samples were diluted with freshly 0.22 μm-filtered PBS before checking the particle sizes and concentrations using the NanoSight NS500 instrument. Five videos of more than 30 second durations each were taken at the camera level of 15 in light scatter mode. All the samples were analysed with the same setting using the NTA 2.3 software.

## Traditional flow cytometry

After the different traffic-light reporter cells were added into EVs at different time points, or after the reporter cells were co-cultured with EV-producing cells for 24 h, GFP expression was quantified using the MACSQuant Analyzer 10 flow cytometer (Miltenyi Biotec, Germany). Briefly, the cells in 96-well plates were washed with PBS once and trypsinized for 5 min at 37 °C. The trypsin was then neutralized using cell medium supplemented with 10% FBS. After adding DAPI to check the cell viability, the cells were sampled by the MACSQuant analyzer using the settings of one specific reporter cell line for all the measurements. The FlowJo software (version 10.6.2) was used to calculate the percentage of GFP positive cells.

## Single-vesicle flow cytometry

HEK293T cells were either transfected with VSV-G-mNeonGreen construct only or co-transfected with VSV-G-mNeonGreen and CD63. Six hours post-transfection, the medium was changed to Opti-MEM medium (Gibco, USA) with 1% Anti-anti. After 2 days, the medium was harvested and centrifuged at 700x $g$ for 5 min, followed by 2000x $g$ for 10 min and subsequently filtered through a 0.22 μm filter system. In a v-bottom 96-well plate, 25 μl of each sample was incubated with APC-labelled CD63 antibody (Miltenyi Biotec, Germany; 1 nM per well) overnight under dark conditions. The samples were then diluted 1000 times and transferred into an R-bottom 96-well plate. Amnis® Cell-Stream instrument (Luminex, US) was utilized to evaluate the engineered EVs at single-vesicle level. The collected data was then analysed using FlowJo software (version 10.6.2).

## EV-addition assays in reporter cells

The reporter cells used in this study were seeded into 96-well plates at the following densities: $1 \times 10^4$ (Hela-TL), $2 \times 10^4$ (T47D-TL), $1.5 \times 10^4$

(B16F10-TL), $5 \times 10^4$ (Raw264.7-TL), $5 \times 10^4$ (THP-1-TL), $5 \times 10^4$ (K562-TL), $1 \times 10^4$ (MSC-TL), $8 \times 10^3$ (HEK-SL) and $2 \times 10^4$ (HEK-Blue-NF-κB) cells per well. The following day, different doses of EVs were added directly into each of the reporter cells, except for HEK-Blue-NF-κB. The doses of the EVs used and the time for incubation are indicated in each figure. GFP positive cells were confirmed either by fluorescent microscopy or by MACSQuant flow cytometry. For HEK-Blue-NF-κB cells, the RS EVs were added directly into the wells for 48 hours and the stimulation (TNF-α, 10 ng/ml) added after another 6 hours. Luciferase signals from the cell lysate were evaluated using the GloMax® 96 Microplate Luminometer machine (Promega, USA).

### Virus production

The transgenes were subcloned into our lentiviral vectors (Transfer plasmid, 22.5 μg/T175 flask) which were co-transfected with pCD/NL-BH (Helper plasmid, 22.5 μg/T175 flask) and pcoPE01 (Envelope plasmid, 3.5 μg/T175 flask) into HEK293T cells and incubated overnight. The next morning, cell medium was changed to complete DMEM medium (with 10% FBS and 1% Anti-anti) supplemented with sodium butyrate (Sigma-Aldrich). After 6 to 8 hours, the sodium butyrate containing DMEM medium was changed back to complete DMEM medium without additional chemicals. Nalgene® Oak Ridge Centrifuge Tubes (Thermo Scientific) were used for harvesting viruses 22 to 24 hours after incubation. Briefly, the virus-particle-containing medium was collected and filtered using a 0.45 μm syringe filter (VWR), and then centrifuged at 25,000 x g for 90 min at 4 °C. The supernatant was aspirated, and freshly prepared medium (IMDM with 20%FBS) was used to resuspend the virus pellets. The viruses were added directly into the target cells or stored at -80 °C for long-term use.

### Stable reporter cell generation

HeLa, T47D, B16F10, Raw264.7, THP-1, K562, and MSC cells were seeded into 6-well plates and the viruses were added into the cells the next day. Titration of viruses was done using 3 doses: 2 μl, 10 μl, and 50 μl per well. After one day incubation of the viruses with target cells, the medium was changed back to normal complete medium (DMEM + 10% FBS + 1% Anti-anti for B16F10 and Raw264.7 cells; RPMI-1640 + 10% FBS + 1% Anti-anti for THP-1, K562, and MSC cells). Two days after virus transduction, the cells were trypsinized and resuspended in fresh medium. Resistance selection was performed by adding puromycin (2 μg/ml for B16F10 and MSC cells, 4 μg/ml for THP-1, K562, HeLa, and T47D cells, and 6 μg/ml for Raw264.7 cells). Untransduced cells died from the puromycin whereas successfully transduced cells survived and continued to grow. The cells were passaged under puromycin selection for approximately one week before the cells were utilized for downstream experiments.

### Direct co-culture of EV-producing cells with reporter cells

HEK293T cells were seeded into a 6-well plate at a density of 0.5 million cells per well. The next day when the cells reached 60-70% confluence, corresponding constructs were transfected into the wells using Lipofectamine2000 (Invitrogen, USA) according to the manufacturer's protocol. Six hours after transfection, the medium was changed to fresh medium (DMEM + 10%FBS + 1% Anti-anti) to reduce the toxicity of the Lipofectamine 2000. Twenty-four hours after plasmid transfection, the cells were trypsinized, counted, and mixed with the corresponding reporter cells at ratios of 1:1 or 1:5, or other ratios as indicated in the Fig.s (ratio=EV-producing cells: reporter cells) in a 96-well plate. After co-culturing for 24 h, the cells were trypsinized and measured using the MACSQuant flow cytometer to check the percentage of GFP positive cells.

### IBIDI co-culture μ-slide assay

HEK293T cells were seeded into a 6-well plate at a density of $5 \times 10^5$ cells per well. The day after, indicated constructs were transfected into

cells using Lipofectamine 2000 (Invitrogen, USA) according to the manufacturer's protocol. To avoid the toxicity of the Lipofectamine2000 on the HEK293T cells, the medium was changed to fresh complete medium (DMEM + 10%FBS + 1% Anti-anti) after 6 h. The following day, the transfected cells were trypsinized and counted. The transfected cells (feeder cells or EV-producing cells) were seeded into the surrounding reservoirs of the ibidi μ-Slide while the recipient cells (traffic-light reporter cells) were added to the central reservoir, following cell numbers indicated in the Fig.s. The volume of medium used for each reservoir was 40 μl. Once the cells were attached to the bottom, another 400 μl of complete medium (DMEM + 10%FBS + 1% Anti-anti) was added into the slide slowly and carefully to immerse the walls between the central reservoir and the surrounding reservoirs such that cell-cell communication could be mediated by engineered EVs. Four days later, the GFP positive cells were measured using either a fluorescent microscope or the MACSQuant flow cytometer.

### Transwell co-culture assay

Similar to the IBIDI assay, HEK293T cells were seeded into a 6-well plate for 24 h and then transfected with the indicated constructs. The medium was changed to fresh complete medium after 6 h of transfection. One day after transfection, the cells were trypsinized and counted, and the EV-producing cells were added to the top chamber of the transwell system (pore size=0.4 μm) while the reporter cells were seeded at the bottom. After 4 days of cell-cell communication by engineered EVs, a fluorescent microscope or MACSQuant flow cytometer was used to check for GFP positive cells.

### Dynamic live imaging assay

Huh7 cells were plated 1 day before the experiment in a polymer-bottom cell culture plate (Ibidi, cat. No. 82426), with $5 \times 10^4$ cells per well. $5 \times 10^{10}$ EVs were added to the cells 3 hours prior imaging and Hoechst dye for nucleus staining was added just before the live cell imaging.

Confocal images were acquired on a Nikon C2 + confocal microscope equipped with an oil-immersion 60x objective with numerical aperture 1.4 (Nikon Instruments, Amsterdam, The Netherlands). The sample was excited and detected with appropriate excitation laser lines and emission filters and the fluorophores were imaged sequentially. The images were taken every hour over the course of 72 hours. The corresponding videos were generated by using the Nikon NIS-Elements Imaging Software.

### ELISA

2 doses ($1 \times 10^9$ and $5 \times 10^8$) of VEDIC and VFIC (Cas9-RNPs as the payload) EVs were lysed in 0.1% TritonX-100 for 30 min in a shaking platform (450 rpm/min). And then the samples were sonicated $4 \times 10$ seconds (incubate on ice for 1 min between bursts). Afterwards, the EV lysates were centrifuged at 14,000 rmp for 5 min at 4 °C. The supernatants were transferred to new tubes and diluted for 5 times with PBS before the ELISA assay (final concentrations with the EV samples were $2 \times 10^8$ /50 μl and $1 \times 10^8$ /50 μl respectively). FastScan™ Cas9 (*S. pyogenes*) ELISA Kit (Cell Signaling Technology, #29666) was ordered to perfom the assay and the main procedure was execurated according to the protocol provided by the manufacturer. Breifly, the standard curve was generated by serial dilution of Cas9 recombinant protein (2 ng/ml; 1 ng/ml; 0.5 ng/ml; 0.25 ng/ml; 0.125 ng/ml and 0 ng /ml). 50 μl of samples (EVs and serial dilution of Cas9 recombinant protein, as well as the positive control samples) were added into the pre-coated 96-well plate (duplicates were run for each sample). And then 50 μl of antibody cocktail (prepared from a mixture of 4× Cas9 rabbit Capture antibody and 4× Cas9 mouse HRP-linked antibody) was added to each well. Seal the plate with the tape and incubate for 1 hour at room temperature on a plate shaker (400 rpm/min). The tape was then gently removed and the plate was washed 3 times with ELISA wash

buffer with 200 µl each time for every well. After that, 100 µl of TMB substrates were added into each well, and seal the tape to incubate in dark for 15 min at room temperature on a plate shaker (400 rpm/min). And then add 100 µl of Stop solution to each well and shake gently for a few seconds. At last, read absorbance at 450 nm within 30 min after adding Stop solution.

## Confocal microscopy
Huh7 cells were seeded in polymer-bottom cell culture plates (Ibidi, cat. no. 82426) one day before the experiment, similar to the dynamic live imaging assay. Indicated number of engineered EVs were added to the cells one day after seeding. After adding EVs for 48 hours, Hoechst dye was added before confocal microscopy imaging. The confocal images were taken the same way as described in dynamic live imaging assay. Image processing was performed by using Fiji software.

## Fluorescent microscopy
After addition of EVs, co-culture, IBIDI, and Transwell assays, the GFP positive cells were visualized under a fluorescent microscope. We chose the area for taking pictures randomly and set up the same parameters for the groups using one experiment. All the images were then processed with the same parameters in the machine or using the Fiji software.

## Western blot analysis
Whole cell protein was isolated using RIPA buffer supplemented with a protease inhibitor cocktail, mixed with sample buffer (4×), and heated at 70 °C for 10 min. For EV samples, $1 \times 10^{10}$ EVs were mixed with sample buffer (4×) and heated at 70 °C for 10 min. Samples were then loaded onto a NuPAGE™ 4-12% Bis-Tris Protein Gel (Thermo Scientific) and ran at 120 V for 2 h in NuPAGE™ MES SDS running buffer (Thermo Scientific). Proteins were transferred from the gel to the membrane using iBlot™ 2 Transfer Stacks (Thermo Scientific). The membrane was blocked with Intercept™ blocking buffer (LI-COR Biosciences) for 1 h at room temperature in a shaker after which it was incubated with primary antibodies overnight at 4 °C. The membrane was washed with TBS-T buffer three times for 5 min each and incubated with corresponding secondary antibodies for 1 h at room temperature in a shaker. After washing with TBS-T buffer three times and with PBS once, the membrane was scanned using the Odyssey infrared imaging system (LI-COR). All the unprocessed blots were provided ethier in the Source Data file (for main figures) or at the end of the supplementary file (for Supplementary Figs.). Detailed information about the primary and secondary antibodies are provided in Supplementary Table 2.

## Gene editing efficiency by Sanger sequencing
N2a cells were seeded into a 24-well plate, with $1 \times 10^5$ cells/well. Then the cells were treated by $1 \times 10^{10}$ Cas9-RNP loaded EVs the following day. 3 days after EV addition, the cells were harvested, and genomic DNA were isolated by using Maxwell® RSC Cell DNA Purification Kit according to the manufacturer's instructions. PCR was performed to amplify the regions flanking the sgRNA targeting site and the PCR products were purified by Monarch DNA Gel Extraction Kit according to the manufacturer's instructions. At last, the purified DNA products were sent for Sanger sequencing and the data were analysed by using online Synthego ICE analysis software. Primer sequences and sgRNA sequence for mTTR were provided in Supplementary Table 5.

## Dynamic imaging of endosomal markers
The cells employed in the endosomal colocalization assay was Huh7 cells, specifically in-house established cell lines overexpressing mRFP-RAB5 and mRFP-RAB7. These stable cell lines were cultured and sustained in DMEM supplemented with 10% fetal bovine serum and 1 µg/ml puromycin to maintian reporter expression. Cultures were maintained at 37 °C with 5% CO2 in 95% humidity using T25 tissue culture

flasks. For experimental procedures, Huh7-Rab5 or Huh7-Rab7 cells were seeded in MatTek glass-bottomed dishes at a density of 30,000 cells per 35 mm dish in complete media (DMEM containing 10% FBS) at least 16 hours before treatment initiation. Nuclei were uniformly stained with Hoechst 33342 (0.25 µg/ml) introduced to the culture medium 30 minutes before imaging experiments. Subsequently, the cell media was replaced with media containing extracellular vesicle (EV) samples, and the Petri dish was promptly transferred to a Nikon CrEST X-Light V3 Spinning Disk microscope equipped with a humidified imaging chamber set at 5% CO2 for continuous imaging over a period of 48 hours.

## IHC staining for p65 expression in the brain
IHC staining for p65 was carried out using a Leica Bond™ system with protocol F30Rb. Mouse brain sections were pre-treated with heat-mediated antigen retrieval using either a citrate-based buffer (epitope retrieval solution 1, pH 6) or an EDTA-based buffer (epitope retrieval solution 2, pH 9) for 20 minutes. Following retrieval, the sections were incubated with the primary p65 antibody (Abcam, #ab16502) at a 1/4000 dilution for 30 minutes at room temperature. Detection was performed using the Bond Polymer Refine Detection Kit (# DS9800), with DAB as the chromogen. The sections were then counterstained with hematoxylin and mounted with Aquatex. The slides were then imaged using the Akoya Biosciences Vectra Polaris imaging system. Detailed information about the primary and secondary antibodies are provided in Supplementary Table 1.

## IHC staining for melanoma tissues
Tissue sections were fixed at 65 °C for 1 hour before the slides were subjected to deparaffinization and rehydration as follows: Xylene for 20 min, 100% ethanol for 3 min twice, 95% ethanol for 3 min, 70% ethanol for 3 min, and then 50% ethanol for 3 min. Afterwards, the slides were rinsed in running cold tap water for 5 min followed by antigen retrieval using citrate buffer, pH 6.0 (Sigma). After antigen retrieval, the slides were washed with PBS three times for 5 min each and immersed in blocking buffer for 30 min at 37 °C. The slides were then incubated with primary anti-GFP antibody (Abcam, ab290, 1:200 dilution) overnight after blocking. The following day, after washing the slides with PBS three times for 5 min each, the slides were incubated with secondary antibody Goat Anti Rabbit IgG H&L (Alexa Fluor® 488) (Abcam, ab150077, 1:500 dilution) for 30 min at 37°C, followed by washing with PBS three times for 5 min each. The slides were mounted using ProLong™ Diamond Antifade Mountant with DAPI (Thermo Scientific) and sealed with nail polish. Images were taken using a confocal microscope (Nikon, Japan). Detailed information about the primary and secondary antibodies are provided in Supplementary Table 1.

## IHC staining for tissues from Cre-LoxP R26-LSL-tdTomato reporter mice
Tissue sections (5 µm) were made from organs of ICV (n = 3 for PBS, and n = 4 for other groups, and in total 15 mice, 12-14 weeks old, gender balanced, female=6 and male=9) and IP (n = 3 each group and 4 groups in total, 12 mice in total, 10-14 weeks old, gender balanced, female=2 and male=10) injected R26-LSL-tdTomato reporter mice (the mice were kept at 20–22 °C, with 40–60% humidity and 14 hours of light/10 hours of night cycle) and then were deparaffinized in xylene and ethanol, boiled in citrate buffer for 20 min, and blocked with 5% goat serum in PBS-T (PBS containing 0.3% Triton X-100) solution for 1 h at room temperature. The sections were then stained with primary antibodies in blocking buffer at 4 °C overnight. After washing with PBS, sections were stained with appropriate fluorophore-conjugated secondary antibodies in PBS or PBS containing 0.1% Triton X-100 for 1 to 2 h before washing and mounting. A Zeiss LSM780 confocal microscope or Zeiss Axioscan Z.1 was used for imaging.

### H&E staining of Hippocampus, cortex and cerebellum

Sagittal brain fixed tissue sections (5 μm) were processed and routinely stained with Hematoxylin-Eosin (H&E). Briefly, the slides were placed in a slide holder and deparaffinized and rehydrated by using xylene, ethanol and $H_2O$. And then the slides were stained by hematoxalin and washed by tap water. Next, the slides were stained by Eosin and dehydrated by ethanol and xylene. Images of different regions of brain were taken by microscope and analysed by a certified pathologist.

### Statistics and reproducibility

Statistical tests for the biological replicates used in this study are reported in Source Data file. legend. R (version 4.4.2) and GraphPad software were utilized for the statistical analysis and the data presented as ±SD. To assess differences between group means, both one-way and two-way analyses of variance (ANOVA) were employed, utilizing the aov() function. The selection between one-way and two-way ANOVA was determined by the experimental design and the number of independent variables. Prior to ANOVA, the assumption of homogeneity of variance was evaluated. When this assumption was satisfied, ANOVA was performed. Significant differences were determined at a significance level of $p < 0.05$. For significant ANOVA results, Tukey's Honestly Significant Difference (HSD) post hoc test, implemented through the TukeyHSD() function, was used to identify specific pairwise group differences. All statistical tests were performed with a 95% confidence. Two-tailed T-test was used for the comparisons of two individual groups. Log-rank (Mantel-Cox) test was used for the survival comparisons. No statistical method was used to predetermine sample size. No data were excluded from the analyses and the experiments were not randomized. The Investigators were not blinded to allocation during experiments and outcome assessment.

### Reporting summary

Further information on research design is available in the Nature Portfolio Reporting Summary linked to this article.

## Data availability

All the data generated in this study are provided in the Supplementary Information/Source Data file. Materials are available upon signing the material transfer agreement (MTA) submitted to S.E.-A. and Evox Therapeutics Limited, Oxford, United Kingdom. Source data are provided with this paper.

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

## Acknowledgements

The confocal imaging was performed at the Live Cell Imaging unit/Nikon Center of Excellence, BioNut, Karolinska Institutet. This study was supported by Cancerfonden (211762 Pj 01 H), the European Research Council (ERC) under the European Union's Horizon 2020 research and innovation programme (DELIVER, grant agreement No 101001374), the European Union's Horizon 2020 research and innovation programme (EXPERT, grant agreement No 825828), the Swedish foundation of Strategic Research (FormulaEx, SM19-0007), Brain foundation (Hjärnfonden) contract (FO2024-0073-TK-113) and the Swedish Research Council (4–258/2021) received by S.E.-A.; by the Swedish Research Council (VR), Grant for half-time research position in a clinical environment (2021-02407), and the CIMED junior investigator grant received by J.Z.N.; by the Swedish Research Council (VR), Grant for half-time research position in a clinical environment (2022-02449), and the CIMED junior investigator grant (FoUI-976434) received by O.W.; CIMED junior investigator grant (FoUI-988637) received by Xiuming Liang; and by Evox Therapeutics Limited.

## Author contributions

Conceptualization: X.L., D.G., J.H., I.M., J.Z.N and S.E.-A. Methodology: X.L., J.X., Z.N., E.V.W., L.V.H., M.G., O.W., W.Z., R.J.W., R.H., D.R.M., J.B., G.Z., H.Z., S.R., H.Y.E., J.R., A.M.Z., A.G., V.W.Q.H., R.S., D.W.H., O.G.J., A.G.U., Y.Z., I.M., C.M.P., T.C.R., and R.E.V. Investigation: X.L., D.G., J.X., Z.N., E.V.W., L.V.H., O.W., W.Z., R.J.W., R.H., D.R.M., J.B., G.Z., H.Z., S.R., A.Z., A.G., D.W.H., O.G.J., A.G.U., Y.Z., C.M.P., T.C.R., D.C., P.V., Z.J.N., R.E.V., and S.E.-A. Visualization: X.L., D.G., J.X., E.K.E., Z.J.N., R.E.V., and S.E.-A. Funding acquisition: Z.J.N., O.W., and S.E.-A. Supervision: S.E.-A. Writing: X.L. Editing: X.L., D.H., O.G.J., T.C.R., Z.J.N., O.W., D.G., A.G.U., A.D.F., M.J.A.W., and S.E.-A.

## Funding

## Competing interests

O.W., J.Z. N., D.G. A.G., M.J.A.W. and S.E.-A. are consultants and stakeholders in Evox Therapeutics Limited, Oxford, United Kingdom. A.D.F., and J.H. are employees of Evox Therapeutics Limited, Oxford, United Kingdom. Evox Therapeutics filed a patent application related to the data used in this work. The patent was applied by Evox Therapeutics Limited. The inventors are Xiuming Liang, Dhanu Gupta, Samir EL Andaloussi and Joel Nordin. Its application number is PCT/EP22/84310 which is still under evaluation. The development of VEDIC and VFIC systems of this work was included in this patent. All the data are available in the manuscript or in the supplemental information. Materials are available upon signing the material transfer agreement (MTA) submitted to S.E.-A. and Evox Therapeutics Limited, Oxford, United Kingdom. The remaining authors declare no competing interests.

## Additional information

[1]Division for Biomolecular and Cellular Medicine, Department of Laboratory Medicine, Karolinska Institutet, Stockholm, Sweden. [2]Karolinska ATMP Center, ANA Futura, Karolinska Institutet, Stockholm, Sweden. [3]Department of Cellular Therapy and Allogeneic Stem Cell Transplantation (CAST), Karolinska University Hospital, Stockholm, Sweden. [4]Cancer Research Laboratory, Shandong University-Karolinska Institutet collaborative Laboratory, School of Basic Medical Science, Shandong University, Jinan, Shandong, PR China. [5]Institute of Developmental and Regenerative Medicine, University of Oxford, IMS-Tetsuya Nakamura Building, Old Road Campus, Roosevelt Dr, Headington, Oxford OX3 7TY, UK. [6]Department of Paediatrics, University of Oxford, South Parks Road, Oxford OX1 3QX, UK. [7]VIB Center for Inflammation Research, VIB, 9052 Ghent, Belgium. [8]Department of Biomedical Molecular Biology, Ghent University, 9052 Ghent, Belgium. [9]Evox Therapeutics Limited, Oxford, United Kingdom. [10]Department of Hepatobiliary Surgery, Shandong Provincial Hospital Affiliated to Shandong First Medical University, Jinan, China. [11]Division of Chemical Biology, Department of Life Sciences, Chalmers University of Technology, 41296 Göteborg, Sweden. [12]Breast Center, Karolinska Comprehensive Cancer Center, Karolinska University Hospital, Stockholm, Sweden. [13]Experimental Cancer Medicine, Clinical Research Center, Department of Laboratory Medicine, Karolinska Institutet., Stockholm, Sweden. [14]Department of Gynecology, The Obstetrics and Gynecology Hospital of Fudan University, 419 Fang-Xie Road, Shanghai 200011, P.R. China. [15]Departamento de Ciencias Basicas, Universidad Industrial de Santander, Bucaramanga, Colombia. [16]Institute for Transfusion Medicine, University Hospital Essen, University of Duisburg-Essen, Essen, Germany. [17]Department of Pharmaceutics, Utrecht Institute for Pharmaceutical Sciences (UIPS), Faculty of Science, Utrecht University, Utrecht 3584 CG, The Netherlands. [18]Department of Medical Biochemistry and Microbiology, Uppsala University, Uppsala, Sweden. [19]Department of Pathology, Shandong Provincial Hospital Affiliated to Shandong First Medical University, Shandong 250021, PR China. [20]Institute of Technology, University of Tartu, 50 411 Tartu, Estonia. [21]MDUK Oxford Neuromuscular Centre, Oxford OX3 7TY, UK. [22]CDL Research, University Medical Center Utrecht, Utrecht, Netherlands. [23]Department of Experimental Cardiology, University Medical Center Utrecht, Utrecht, Netherlands. [24]Department of Clinical Immunology and Transfusion Medicine (KITM), Karolinska University Hospital, Stockholm, Sweden. [25]These authors contributed equally: Dhanu Gupta, Junhua Xie.
✉e-mail: xiuming.liang@ki.se; joel.nordin@ki.se; samir.el-andaloussi@ki.se

