## [Transparent Peer Review File · Nature Communications]

Engineering of extracellular vesicles for efficient intracellular delivery of multimodal therapeutics including genome editors

Corresponding Author: Dr Xiuming Liang

This manuscript has been previously reviewed at another journal. This document only contains information relating to versions considered at Nature Communications. Mentions of the other journal have been redacted.

Version 0:

Reviewer comments:

Reviewer #1

(Remarks to the Author)

The authors demonstrated in this manuscript that 1) free-form protein payloads can be loaded into the EV via introduction of an engineered mini-intein protein with self-cleavage activity (especially in low pH conditions) and 2) introduction of VSV-G, a viral fusogen, can enhance both endosome escape of uptaken exosomes and target cell engagement. They further validated this idea using various functional protein payloads including Cre recombinase, Cas9 and an NF- κ B inhibitor super-repressor I κ B in vitro and in vivo. Even though some issues have been adequately responded by the authors, there are still several issues to be answered.

Major concerns:

1. The main novelty of current study comes from the observation that engineering of VSV-G (whether in conjunction with cargo loading mechanism or independently expressed on the surface of EVs) can dramatically enhance intracellular delivery of functional payloads presumably enhancing via endosomal escape and partly by enhancing cellular uptake. Even though VSV-G has long been known for its fusogenic activity thus has been utilized for enhancing exosome-based payload delivery (Kim et al, Sci Adv 2020; Yang et al., Adv Mater, 2017; Ilahibaks et al, J Control Release 2023). Even though the authors showed systematic analysis showing superiority of VSV-G compared to other potential fusogens, the current study lacks substantial novelty. Especially, Ilahibaks's group has utilized almost same scheme to use VSV-G as cargo loading and delivery, named TOP-EV in their 2023 paper.

2. Cleavage-dependent mechanisms have also been shown effective for free-form cargo loading and delivery. Usage of intein provides a marginal advantage (no effort to manipulate for inducing cleavage in the target cells) over previously reported photo-cleavable mechanisms. However, almost all data showing superiority of VEDIC or VFIC-EV didn't come from this novel loading mechanism but from the usage of VSV-G.

Minor Issues:

1. EVs are isolated only by ultrafiltration (TFF and amicon spin filter). These methods are excellent for concentration of EVs but obvious limitation for the inability to remove any macromolecular contaminants. It is recommended to include chromatographic purification step for reliable purification of EVs.

2. The authors insisted to use PEI-based transient transfection to obtain their engineered EVs. Since several transient transfection methods can interfere (enhance) exosome-based uptake and cargo delivery efficiency, two alternatives can be recommended. If PEI-based transfection methods are to be used, the EVs should be produced from at least pooled clones (stable cells). Or viral transduction (which have been used to generate Cre reporter cell lines for the manuscript) should be used to produce EVs after transient infection.

3. As the authors indicated, the prolonged expression of VSV-G usually induce cell fusion and eventual cell death. Therefore, there are two issues related to this notion. First, cell viability should be carefully checked after expression of VSV-G. The higher expression of VSVG, the higher ratio of cell death can be expected. Secondly, potentially apoptotic bodies for dying cells can be included in the VSV-G engineered cell-derived EV fractions. Furthermore, dying cells (not necessarily apoptotic marker positive yet) can produce EVs containing quite different payloads compared to healthy cells. In case of Cre or Cas9 experiments, these factors would not affect the results; however, in case of delivering anti-inflammatory payloads (such as super-repressor I κ B shown in revised figure 7) the potential changes in exosome components due to cell stress (induced by VSV-G overexpression) might have huge functional impact. Since overexpression of VSV-G definitely induce cellular stresses, the impact of VSV-G expression itself should be tested in time- and dose-dependent manner.

Reviewer #3

(Remarks to the Author)

My previous concerns have been addressed

Reviewer #4

(Remarks to the Author)

The authors demonstrated exciting new concept to improve delivery with payload that its release is dependent on cleavage mechanism. It's an exciting discovery. Reviewer 2 had many great questions and authors answered with data very nicely for the most questions.

1) Q6: Unable to detect Cas9 with Cas9 antibody is questionable even though VSV-G is very well detected
2) Q7: I have to agree with the reviewer 2 that the images do not seem to match with the quantification. Additionally, flow cytometry quantification that the authors used is questionable. It's occasionally conducted and claimed that various cell types in brain can be analyzed by flow cytometry or quantified in nucleus level. However, method doesn't provide enough details and flow chart doesn't show as clear separation of cells to clearly claim that gating was conducted perfectly. Overall, I believe that it's fairly challenging to accept the quantification. I would believe that even DNA quantification of edited DNA will be more accurate than this method.

Even if it is decided to be published, it will be better to clearly state that it's quantified by flow cytometry in figure legend.

Overall, authors did great job.

Reviewer #5

(Remarks to the Author)

The authors describe an engineered EV approach to enhance the intracellular delivery of therapeutic proteins. Critical to their designs are the inclusion of an intein linker which provides pH-sensitive release of the therapeutic cargo, and VSVg fusogen which mediates endosomal escape and cytosolic release of the protein cargo. They demonstrate the utility of this technology via two platforms, VSVg plus CD63 EV-sorting Domain-Intein-cargo (VEDIC) and VSVg-Foldon-Intein Cargo (VFIC). The authors demonstrate the ability of both of these platforms to deliver proteins with direct therapeutic implications, such as PCSK9 editors and NF- κ B super repressors, as well as enable delivery to difficult targets, such as cells of the CNS. Overall, the manuscript is well written and the described experiments are designed and executed well. However, there are a number of points that, if addressed, would enhance the quality of the publication and further assert the author's claims.

- The importance of the VSVg fusogen is clear, however given the established utility of VSVg VLPs and vesicles for intracellular protein delivery, the novelty of the studies outlined in this manuscript comes into question. No doubt, the approach to incorporate VSVg alongside the intein system into the EV design is conceptually compelling and well executed, however the advantage of the VEDIC and VFIC modalities over other established VSVg VLPs is unknown. Functional and quantified intracellular protein delivery comparisons to other VSVg-based protein delivery vehicles could emphasize the synergy of VSVg+intein and highlight the utility of the VEDIC and VFIC platforms.
- When compared head-to-head, VFIC demonstrates superior potency across various cargos both in vitro and in vivo, aside for CNS applications. It is unclear whether this is due to some undefined and CNS-specific advantage unique to the VEDIC platform in comparison to the VFIC platform. The authors demonstrate that Cre recombination primarily occurred in astrocytes and microglial cells. It is unclear if these cells, which appear to be more permissive to protein delivery, are more actively recruited to the infusion site by VEDICs in comparison to VFICs thus enabling a greater level of delivery/editing to be capable with the VEDICs. The authors should provide some quantitative data to account for differences in astrocyte and microglial recruitment between VEDIC and VFIC treated mice.
- The authors should provide additional citations that support the use and translational/clinical relevance of osmotic pumps for CNS delivery of therapeutics.
- In terms of EV QC, are there any empty EVs in the isolated populations? This would be important to know since mice are dosed based on the total number of EVs and significant differences in the ratio of loaded/unloaded EVs could account for differences in potency across VEDIC and VFIC.
- Does the use of CD63 make the VEDIC system more susceptible to recognition by immune cells, as seen in the enhanced delivery to the spleen? Additionally, would it effect the redosability of the EVs? Does potency decline upon repeated administration?
- For systemic LPS-induced ARDS models, the lung is typically a critical organ that is looked at, especially with regards to changes in the protein content of BALF. Is there a reason why the lungs were not examined in these studies?

Version 1:

Reviewer comments:

Reviewer #1

(Remarks to the Author)

My previous concerns have been adequately addressed.

Reviewer #4

(Remarks to the Author)

There are no additional comments

Reviewer #6

(Remarks to the Author)

I have carefully reviewed the main manuscript and response letter from the authors. There is no doubt that the authors did a great work for facilitating EVs for protein and RNA therapeutic delivery in vitro and in vivo. There are several comments need to be addressed before its publication.

1. Paragraph 1 (introduction part), it is not so accurate that LNP has limitation related to endosomal entrapment and toxicity for protein delivery, there are a lot of work focused on LNP for protein delivery. Here should be reword.
2. Paragraph 2, not sure how "synthetic properties induce various side effects when they are applied in vivo". Why? LNP, polymers, CPPs are not generated in vivo. Please re-word here.
3. For the preparation of VEDIC system, how many proteins can be encapsulated into the EVs?
4. A quick question about the intratumoral injection part. If the authors want to show the in vivo Cre delivery performance by EVs, using Ai9 or similar mice model is ok, rather than using tumor-bearing mice.
5. Could the authors explain why harvesting the mice after 1 week injection for conducting IF? Moreover, a subsequent flow cytometry should be added along with the IF imaging (Figure 6).
6. I am curious about the inflammatory cytokines level before and after EVs treatment on LPS-induced inflammation mice models. Both in the serum and BALF.
7. what is the safety level when using high dose of EVs for the delivery of cargoes? Could the authors provide H&E staining of the main organs after high dosing of EVs?

Version 2:

Reviewer comments:

Reviewer #6

(Remarks to the Author)

The authors addressed my comments well. I recommend its publication.

Reviewer #1 (Remarks to the Author):

The authors demonstrated in this manuscript that 1) free-form protein payloads can be loaded into the EV via introduction of an engineered mini-intein protein with self-cleavage activity (especially in low pH conditions) and 2) introduction of VSV-G, a viral fusogen, can enhance both endosome escape of uptaken exosomes and target cell engagement. They further validated this idea using various functional protein payloads including Cre recombinase, Cas9 and an NF- κ B inhibitor super-repressor I κ B in vitro and in vivo. Even though some issues have been adequately responded by the authors, there are still several issues to be answered.

Major concerns:

1. The main novelty of current study comes from the observation that engineering of VSV-G (whether in conjunction with cargo loading mechanism or independently expressed on the surface of EVs) can dramatically enhance intracellular delivery of functional payloads presumably enhancing via endosomal escape and partly by enhancing cellular uptake. Even though VSV-G has long been known for its fusogenic activity thus has been utilized for enhancing exosome-based payload delivery (Kim et al, Sci Adv 2020; Yang et al., Adv Mater, 2017; Ilahibaks et al, J Control Release 2023). Even though the authors showed systematic analysis showing superiority of VSV-G compared to other potential fusogens, the current study lacks substantial novelty. Especially, Ilahibaks's group has utilized almost same scheme to use VSV-G as cargo loading and delivery, named TOP-EV in their 2023 paper.

Response: Thank you for pointing out this novelty concern. **We have here for the first time utilized a self-cleavage protein, intein, tethered to EV/sorting domains to realise the enrichment and subsequent release of the cargoes inside of EVs.** This is completely endogenous and modular, and does not require any exogenous reagents, such as a ligand to induce dimerization used in TOP-EVs. Our strategy is different from the TOP-EV you mentioned in several regards:

1. The approach for loading cargoes is different. TOP-EVs exploited an exogenous drug to tether the target protein to an EV-sorting domain (N-Myr). Therefore, 3 components are indispensable for the system to work: Rapamycin (ligand), VSV-G and Target protein-EV-sorting domain expression plasmid. However, we harnessed EV-sorting domain directly fused to the target protein, with the self-cleavage protein, intein, in between. For our VEDIC system, 2 components are required: VSV-G and EV-sorting domain-Intein-Cargo plasmids. In terms of the VFIC system, only one plasmid is needed to efficiently deliver, in theory, any protein of interest (**using VSV-G as both EV-sorting domain and fusogenic protein is novel too**). Therefore, our system is simpler and can be easily engineered to achieve functional protein delivery intracellularly.

2. It is not clear how the protein cargoes can be soluble (release from EV-sorting domain) inside of the TOP-EVs since the ligand is always there. However, for our VEDIC and VFIC systems, adequate data has been provided to demonstrate that the pH-dependent intein will release the cargo proteins during EV-biogenesis, leading to the soluble form of proteins inside of EVs that facilitate their release in recipient cells.

3. Even though we did not compare the efficiency of TOP-EVs with our system, from the published data in the paper, our engineered EVs appear more potent. In T47D reporter cells, 1×10^9 dose of TOP-EVs achieved around 10% recombination while the same dose of our

engineered EVs generate 90% of GFP positive cells (**Figure 4B** in the TOP-EVs paper and **Figure 1g** in our revised manuscript). In terms of CRISPR/Cas9-RNP delivery, our VFIC EVs induced nearly 80% editing HEK-SL cells whilst TOP-EVs achieved less than 30% editing in the same reporter cells (**Figure 6H** in the TOP-EVs paper and **Figure 5d** in our revised manuscript).

Figure Redacted

2. Cleavage-dependent mechanisms have also been shown effective for free-form cargo loading and delivery. Usage of intein provides a marginal advantage (no effort to manipulate for inducing cleavage in the target cells) over previously reported photo-cleavable mechanisms. However, almost all data showing superiority of VEDIC or VFIC-EV didn't come from this novel loading mechanism but from the usage of VSV-G.

Response: The photo-cleavable system is another strategy to make the target proteins soluble inside of engineered EVs. However, it needs exogenous light, such as UV or blue lights (which may negatively affect the EV-producing cells or the isolated EVs), to separate the cargoes. The aim of our study was to identify an endogenous mechanism to separate the cargoes inside of EVs during EV-biogenesis. We did not compare our intein strategy to the photo-cleavable strategy directly because the photo-cleavable system has not been established in our lab. What we would like to emphasize is that the intein used in our study is another novel strategy to efficiently separate cargoes inside of EVs, and together with EV-sorting domain and VSV-G, the developed VEDIC and VFIC systems are efficient for the delivery of functional cargoes both in vitro and in vivo. We agree with you that VSV-G is a very important component of both the VEDIC and VFIC systems, but other components, such as the intein and EV-sorting domains are crucial as well because CD63-Cre+VSV-G (without intein) and intein-Cre+VSV-G (without EV-sorting domain) showed background level of recombination in reporter cells (Revised manuscript **Figure 1g**).

Revised manuscript Figure 1g.

Minor Issues:

1. EVs are isolated only by ultrafiltration (TFF and amicon spin filter). These methods are excellent for concentration of EVs but obvious limitation for the inability to remove any

Editorial Note: Supplementary Figure 2f in this Peer Review File is reproduced with permission from BioRender, *Created in BioRender. Zheng, W. (2025) https://BioRender.com/v11h404*

macromolecular contaminants. It is recommended to include chromatographic purification step for reliable purification of EVs.

Response: Thank you for this good comment and we completely agree. Therefore, we used another 2 methods to purify VFIC EVs: TFF with size exclusion chromatography (SEC) or density gradient ultracentrifugation (DGUC). Using both methods, the activity of isolated EVs was preserved (Revised manuscript **Figure 2i and 2j**, and **supplementary Figure 2f and 11e**). TFF plus SEC was also used to purify the VEDIC EVs and the potency was comparable to that of EVs isolated by only TFF (Revised manuscript **supplementary Figure 2g**). These results indicate that the VEDIC and VFIC EVs are efficient to deliver cargoes regardless of the isolation methods exploited in the study.

Revised manuscript Figure 2i and 2j

Revised manuscript supplementary Figure 2f

Revised manuscript supplementary Figure 11e

Revised manuscript supplementary Figure 2g

2. The authors insisted to use PEI-based transient transfection to obtain their engineered EVs. Since several transient transfection methods can interfere (enhance) exosome-based uptake and cargo delivery efficiency, two alternatives can be recommended. If PEI-based transfection methods are to be used, the EVs should be produced from at least pooled clones (stable cells). Or viral transduction (which have been used to generate Cre reporter cell lines for the manuscript) should be used to produce EVs after transient infection.

Response: Thank you for this advice and suggesting two alternatives. Early studies showed that the generation of stable packaging cell lines for pseudotyped retroviral or lentiviral vectors has been difficult because of the toxicity resulting from chronic expression of VSV-G in most mammalian cells (Burns, et al. Vesicular stomatitis virus G glycoprotein pseudotyped retroviral vectors: Concentration to very high titer and efficient gene transfer into mammalian and nonmammalian cells. *Nati. Acad. Sci. USA*, 1993; Yee, et al. A general method for the generation of high-titer, pantropic retroviral vectors: Highly efficient infection of primary hepatocytes. *Proc. Nati. Acad. Sci. USA*, 1994). We tried to generate the VSV-G stable cells but without success. We do not think viral transduction is a feasible method for us to produce such big amount of engineered EVs used in this study, especially for animal work. Transient transfection using PEI is efficient and the EV production can be scalable. In addition, transient transfection is one of the common methods used by other laboratories as well to produce engineered vesicles, such as the eVLPs produced by David R Liu's lab after transient transfection of 4 plasmids using Polyplus (Banskota, et al. Engineered virus-like particles for efficient in vivo delivery of therapeutic proteins. *Cell*, 2021). Similarly, many AAVs and other viral vectors that are used clinically are produced from HEK cells transiently transfected with PEI.

3. As the authors indicated, the prolonged expression of VSV-G usually induce cell fusion and eventual cell death. Therefore, there are two issues related to this notion. First, cell viability should be carefully checked after expression of VSV-G. The higher expression of VSVG, the higher ratio of cell death can be expected. Secondly, potentially apoptotic bodies for dying cells can be included in the VSV-G engineered cell-derived EV fractions. Furthermore, dying cells (not necessarily apoptotic marker positive yet) can produce EVs containing quite different payloads compared to healthy cells. In case of Cre or Cas9 experiments, these factors would not affect the results; however, in case of delivering anti-inflammatory payloads (such as super-repressor IκB shown in revised figure 7) the potential changes in exosome components due to cell stress (induced by VSV-G overexpression) might have huge functional impact. Since overexpression of VSV-G definitely induce cellular stresses, the impact of VSV-G expression itself should be tested in time- and dose-dependent manner.

Response: Thank you for these comments. Yes, prolonged expression of VSV-G usually induces toxicity to the cells. Cell fusion after VSV-G transfection is rare, which can happen only when the pH drops in the cell culturing condition (VSVG fuses with endosomal membranes at low pH, not at the plasma membrane at neutral pH). For our VSV-G engineered EVs, we transfected the producing cells with VSV-G for 48 hours only and we did not expect to see severe cell toxicity after this short period of time. To corroborate our assumption, and as requested, we performed the time and dose dependent VSV-G transfection into HEK-293T cells and monitored cell viability and morphology changes. As shown in **Evidence Figure 1**, compared to the empty vector, the cell viability (around 90%) did not decrease for VSV-G transfection at the indicated time points after transfection. The cell morphology appeared similar between empty vector and VSV-G transfected cells as demonstrated in **Evidence Figure 2**. In addition, we transfected increasing doses of VSV-G to the cells and did not observe any reduced cell viability after transfection for 72 hours (**Evidence Figure 3**). These results suggest no severe cell toxicity after short-term transient transfection of VSV-G.

We agree with the reviewer that overexpression of VSV-G can induce cellular stresses and this may result in differential loading of payloads into produced EVs. This is an interesting point and we may explore this further in future studies. However, in the current study, the main content is about Cre delivery both in vitro and in vivo, with Cas9 and super-repressor I κ B just as examples to further confirm the delivery efficiency of our engineered EVs. There are a range of lentiviruses used clinically (for CAR/T cell generation as an example) and VLPs that are soon entering clinical trials. These are all produced from cells transiently transfected with VSVG plasmids.

Evidence Figure 1

Evidence Figure 2

Evidence Figure 3

Reviewer #3 (Remarks to the Author):

My previous concerns have been addressed

Reviewer #4 (Remarks to the Author):

The authors demonstrated exciting new concept to improve delivery with payload that its release is dependent on cleavage mechanism. It's an exciting discovery. Reviewer 2 had many great questions and authors answered with data very nicely for the most questions.

1) Q6: Unable to detect Cas9 with Cas9 antibody is questionable even though VSV-G is very well detected

Response: Thank you for this comment. We ordered another batch of Cas9 antibody from abcam (#ab191468) and repeated this experiment. Unfortunately, we again did not pick up the signal of Cas9, but the pattern for VSV-G protein dynamic change is consistent with the previous results. The uptake of VSV-G engineered EVs increased gradually from 6 h to 48 h and peaked at 48 h, degrading quickly at 72 h time point (**Evidence Figure 4** and Revised manuscript **supplementary Figure 11c**). The sensitivity of primary antibody is key for western blot and we assume the commercial Cas9 antibodies are not sensitive enough to detect low level of Cas9 proteins in the cells (we assume only a small amount of engineered EVs would be taken up by cells). The sensitivity of VSV-G antibody is good enough to see VSV-G protein level in a dynamic manner, and VSV-G protein level could indirectly reflect the dynamic change of cargo level in recipient cells.

Evidence Figure 4

C

Revised manuscript supplementary Figure 11c

2) Q7: I have to agree with the reviewer 2 that the images do not seem to match with the quantification. Additionally, flow cytometry quantification that the authors used is questionable. It's occasionally conducted and claimed that various cell types in brain can be analyzed by flow cytometry or quantified in nucleus level. However, method doesn't provide enough details and flow chart doesn't show as clear separation of cells to clearly claim that gating was conducted perfectly. Overall, I believe that it's fairly challenging to accept the quantification. I would believe that even DNA quantification of edited DNA will be more accurate than this method.

Response: Thank you for these nice comments. Regarding the mismatch between images and quantification, we assume there is some misunderstandings. The images are from the ICV injection while the quantifications are from osmotic pump ICV delivery, the latter generating significantly more recombined cells. In respect to the flow cytometry quantification, we added a more detailed explanation to describe how to separate single cells from the mice brains in the methods part of the revised manuscript (**Flow cytometry of single cells from brain tissues**).

(The terminal sedation of mice was performed with I.P administration of Ket/Xyl (Ket/Xyl mix=100 mg/kg ketamine; 20 mg/kg xylazine). And then blood was collected from the right ventricle, and PBS/heparin was used for trans cardinal perfusion (10 ml per mouse at rate of 3 ml/min). Next, the brain samples were collected in 400 μ l serum free RPMI medium at 4°C, followed by adding 200 400 μ l enzyme mix (Enzyme mix = 10 U/mL collagenase I, 400 U/mL collagenase IV, and 30 U/mL DNaseI (Worthington) in PBS). In the next step, the brain samples were cut into small pieces with scissor, and incubated for 10 min at 37°C in oven without shaking. Pipette up and down the samples and incubate 10 min at 37°C in oven without shaking, and repeat this for another 2 times. And then put the single cell mixture on a cell strainer (70 μ m, Falcon) and collect cells into a new Eppendorf 1.5 ml tubes for pellet. Rinse the tubes with 500 μ l RPMI medium and put this also onto the cell strainer, and spin down the cells at 400 g for 7 min at 4°C. Resuspend the cells in 40% percoll in HBSS (2 ml solution=1.2 ml HBSS+0.8 ml percoll). And then spin down the cells at 600 g for 10 min at 4°C without breaks. Resuspend the cell pellet in staining buffer and spin down the cells again at 600 g for 10 min at 4°C. At last, resuspend the cell pellet in PBS and let the cells go through the cell strainer again before flow cytometry. Run 500,000 cells for each sample on the flow and the data was analyzed by using FlowJo.)

Regarding the separation of cells shown in the flow chart, we analyzed 500,000 cells in the flow cytometry assay and gated the majority of cells to avoid bias during analysis. If we gated small amount of the cells, such as 5% of the total shown below (**Evidence Figure 5**), the separation is very clear and the editing effect is even higher. However, even though the populations look much better, gating this way is biased. Thus, we would like to keep the gating strategy demonstrated in the manuscript.

In addition, we repeated the experiment using combination of different EV-sorting domains identified in our lab and the editing efficiency was even higher than the engineered EVs shown in our manuscript (**Evidence Figure 6**). Therefore, we are confident with the flow cytometry data from the osmotic pump ICV injection.

Evidence Figure 5

Evidence Figure 6

Even if it is decided to be published, it will be better to clearly state that it's quantified by flow cytometry in figure legend.

Response: Thank you for the suggestion, and we have now added this to the figure legend in the revised manuscript.

Overall, authors did great job.

Reviewer #5 (Remarks to the Author):

The authors describe an engineered EV approach to enhance the intracellular delivery of therapeutic proteins. Critical to their designs are the inclusion of an intein linker which provides pH-sensitive release of the therapeutic cargo, and VSVg fusogen which mediates endosomal escape and cytosolic release of the protein cargo. They demonstrate the utility of this technology via two platforms, VSVg plus CD63 EV-sorting Domain-Intein-cargo (VEDIC) and VSVg-Foldon-Intein Cargo (VFIC). The authors demonstrate the ability of both of these platforms to deliver proteins with direct therapeutic implications, such as PCSK9 editors and NF- κ B super repressors, as well as enable delivery to difficult targets, such as cells of the CNS. Overall, the manuscript is well written and the described experiments are designed and executed well. However, there are a number of points that, if addressed, would enhance the quality of the publication and further assert the author's claims.

- The importance of the VSVg fusogen is clear, however given the established utility of VSVg VLPs and vesicles for intracellular protein delivery, the novelty of the studies outlined in this manuscript comes into question. No doubt, the approach to incorporate VSVg alongside the intein system into the EV design is conceptually compelling and well executed, however the advantage of the VEDIC and VFIC modalities over other established VSVg

VLPs is unknown. Functional and quantified intracellular protein delivery comparisons to other VSVg-based protein delivery vehicles could emphasize the synergy of VSVg+intein and the highlight the utility of the VEDIC and VFIC platforms.

Response: Thank you for this nice suggestion. The established VSVg VLP system utilizes a viral cleavable peptide (GAG/POL) as an approach to make the protein cargoes soluble inside of the vesicles, thus having a similar function as the self-cleavage intein used in our study. Our goal was to use as few viral components as possible to achieve efficient delivery of cargoes. In response to the comment from the reviewer we have now compared our VEDIC system with the published Nanoblade VLP system (Mangeot, et al. Genome editing in primary cells and in vivo using viral-derived Nanoblades loaded with Cas9-sgRNA ribonucleoproteins. Nature communications, 2019) to deliver Cre recombinase and found VEDIC was as good as Nanoblade in T47D-TL cells, but much better than Nanoblade in HeLa-TL and B16F10-TL cells (Revised Manuscript **Figure 1k**), indicating the utility of the platforms developed in our laboratory.

Revised Manuscript Figure 1k

- When compared head-to-head, VFIC demonstrates superior potency across various cargoes both in vitro and in vivo, aside for CNS applications. It is unclear whether this is due to some undefined and CNS-specific advantage unique to the VEDIC platform in comparison to the VFIC platform. The authors demonstrate that Cre recombination primarily occurred in astrocytes and microglial cells. It is unclear if these cells, which appear to be more permissible to protein delivery, are more actively recruited to the infusion site by VEDICs in comparison to VFICs thus enabling a greater level of delivery/editing to be capable with the VEDICs. The authors should provide some quantitative data to account for differences in astrocyte and microglial recruitment between VEDIC and VFIC treated mice.

Response: Thank you for this nice suggestion. We have now repeated the ICV injection of the engineered EVs with the same settings as shown in the manuscript and co-stained the whole brain section for tdTomato and GFAP. We did not see big difference in astrocyte (GFAP positive) recruitment around the infusion site between VEDIC and VFIC treated mice (**Evidence Figure 7**), the astrocyte distribution pattern of which is similar to the PBS group. At the same time, IBA1 (microglia marker) staining showed no big difference of microglia recruitment surrounding the infusion sites between VEDIC and VFIC groups (**Evidence Figure 8**).

tdT+/GFAP

PBS

CD63-Intein-Cre+VSV-G

VSV-G-Foldon-Intein-Cre

Evidence Figure 7

tdT+/IBA1

PBS

CD63-Intein-Cre+VSV-G

VSV-G-Foldon-Intein-Cre

Evidence Figure 8

- The authors should provide additional citations that support the use and translational/clinical relevance of osmotic pumps for CNS delivery of therapeutics.

Response: Thank you for this good suggestion. We added 2 osmotic pumps-related references that support their applications and translational relevance in our revised manuscript (White, et al. Using Osmotic Minipumps for Intracranial Delivery of Amino Acids and Peptides. *Methods in Neurosciences*, 1994; Wang, et al. Osmotic Pump-based Drug-delivery for In Vivo Remyelination Research on the Central Nervous System. *Journal of Visualized Experiments*, 2021).

- In terms of EV QC, are there any empty EVs in the isolated populations? This would be important to know since mice are dosed based on the total number of EVs and significant differences in the ratio of loaded/unloaded EVs could account for differences in potency across VEDIC and VFIC.

Response: Thank you for this comment. Yes, we definitely think there are empty EVs (non-engineered EVs) in the isolated populations. To define the percentage of engineered EVs for VEDIC and VFIC systems, we performed single vesicle flow cytometry by co-staining for VSV-G (VSV-G-AF488 FITC) and CD63 (CD63-APC), or co-staining for VSV-G (VSV-G-AF488 FITC) and PAN (PAN-APC, CD9/CD63/CD81 together, which supposedly stain the entire population of EVs) (**Evidence Figure 9**). After calculation, around 25% of the VFIC EVs were engineered as demonstrated by percentage of VSV-G+ EVs out of PAN+ (CD9+/CD63+/CD81+) population while around 55% of the VEDIC EVs were engineered as indicated by percentage of CD63+/VSV-G+ EVs out of PAN+ (CD9+/CD63+/CD81+) population (**Evidence Figure 10**).

DP: double positive.

Evidence Figure 9

Evidence Figure 10

We dosed the mice based on the total number of EVs, but not the payloads, because we think this would be a fairer approach since the engineering capacity or loading/enrichment capacity

of the systems is one of the important factors that determine the subsequent delivery efficiency. Normally, the more engineered EVs in the population, the more efficient the system will be.

- Does the use of CD63 make the VEDIC system more susceptible to recognition by immune cells, as seen in the enhanced delivery to the spleen? Additionally, would it effect the redosability of the EVs? Does potency decline upon repeated administration?

Response: Thank you for these comments. With respect to your question whether CD63 makes the VEDIC system more susceptible to the recognition by immune cells, we are not sure. However, there is one publication showing that CD63+ exosomes mediated the transfer of miRNAs from T cells to antigen-presenting cells (APCs) (Mittelbrunn, et al. unidirectional transfer of microRNA-loaded exosomes from T cells to antigen-presenting cells. *Nature communications*, 2011). Therefore, CD63 may play a role in the recognition and subsequent uptake by immune cells. Having that said, we have never seen any differences in PK or PD of EVs engineered with different scaffolds (CD63, CD9, CD81, or Lamp2). Thus, it is unlikely that CD63 EVs behave differently from other EVs. We do not know exactly if CD63 would affect the redosability of the EVs, but we would say re-dosing may be challenging due to the production of neutralization antibodies against VSV-G after first dosing. That's why we are now focusing on the delivery of gene editing tools, such as Cas9-RNPs or base editor, to achieve once-and-done delivery to have permanent change in the cell genome.

- For systemic LPS-induced ARDS models, the lung is typically a critical organ that is looked at, especially with regards to changes in the protein content of BALF. Is there a reason why the lungs were not examined in these studies?

Response: Thank you for pointing this out. We kept the embedded lung tissues from the experiment, but did not keep the BALF as you mentioned. Therefore, we evaluated the tissue damages in lung after LPS induction and found protective role of our engineered EVs (CD63-Intein-SR+VSV-G and VSV-G-Foldon-Intein-SR) as indicated by less inflammatory cell infiltration in the tissues. We have now included the data in the Revised manuscript **supplementary Figure 16e**.

Revised Manuscript supplementary Figure 16e. Yellow arrows indicated infiltrated inflammatory cells in lung tissues.

Reviewer #6 (Remarks to the Author):

I have carefully reviewed the main manuscript and response letter from the authors. There is no doubt that the authors did a great work for facilitating EVs for protein and RNA therapeutic delivery *in vitro* and *in vivo*. There are several comments need to be addressed before its publication.

1. Paragraph 1 (introduction part), it is not so accurate that LNP has limitation related to endosomal entrapment and toxicity for protein delivery, there are a lot of work focused on LNP for protein delivery. Here should be reword.

Response: Thank you for pointing this out. We have modified the expression to a more neutral tone as follows: “the potential limitations related to endosomal entrapment and toxicity have been reported.”

2. Paragraph 2, not sure how “synthetic properties induce various side effects when they are applied *in vivo*”. Why? LNP, polymers, CPPs are not generated *in vivo*. Please re-word here.

Response: Thank you for this comment. LNPs, polymers, and CPPs, as you mentioned, are not naturally produced *in vivo* and are foreign to the body, which raises potential concerns about immunogenicity. We have adjusted the wording to soften the statement as follows: “Another drawback of these strategies is that their synthetic nature may cause various side effects when used *in vivo*.”

3. For the preparation of VEDIC system, how many proteins can be encapsulated into the EVs?

Response: We appreciate this comment and performed ELISA to quantify Cas9 protein levels in the engineered EVs, as no commercial ELISA kit is available for Cre recombinase detection. Our results showed approximately 0.34 Cas9 molecules per engineered VEDIC EV (CD63-Intein-Cas9+VSV-G+sgRNA) and around 1.3 Cas9 molecules per VFIC EV (VSV-G-Foldon-Intein-Cas9+sgRNA). This difference may partially explain the enhanced Cas9-RNP delivery observed with the VFIC system. We have included this data in **Fig. 5f** of the main manuscript.

f

Fig. 5f: Quantification of Cas9 proteins in the engineered VEDIC and VFIC EVs.

4. A quick question about the intratumoral injection part. If the authors want to show the in vivo Cre delivery performance by EVs, using Ai9 or similar mice model is ok, rather than using tumor-bearing mice.

Response: Thank you for this comment. We agree that the Ai9 mouse model is well-suited for demonstrating in vivo Cre delivery. However, we chose tumor-bearing mice to highlight the therapeutic potential of our engineered EVs for cancer treatment. By leveraging VEDIC EVs, we may simultaneously deliver multiple tumor suppressors to tumor tissue, thereby potentially inhibiting tumor growth. This approach expands the potential applications of EV-based therapies in the future.

5. Could the authors explain why harvesting the mice after 1 week injection for conducting IF? Moreover, a subsequent flow cytometry should be added along with the IF imaging (Figure 6).

Response: Thank you for this comment. There is no specific reason for selecting the 1-week time point. However, we assume that one week is sufficient to observe editing effects in the brain, given that our in vitro experiments showed robust editing as early as 24 hours after adding the engineered EVs (**Manuscript Fig. 1i**). This suggests that Cre-mediated recombination occurs relatively quickly.

Fig. 1i: Robust Cre-mediated recombination by engineered EVs was observed 24 hours after the addition of VEDIC EVs.

We appreciate the suggestion to include flow cytometry alongside IF imaging. However, we believe it may not be feasible to detect significant signals via flow cytometry, as tdTomato expression was observed only in limited regions of the brain following ICV injection. This is why we applied osmotic minipumps to raise editing levels. This was quantified by flow cytometry as indicated in **Fig. 6d** of the main manuscript.

Fig. 6d: Percentage of tdTomato⁺ cells in the brain tissues after osmotic pump ICV injection of engineered EVs quantified by flow cytometry, analyzed 4 days after the infusion.

6. I am curious about the inflammatory cytokines level before and after EVs treatment on LPS-induced inflammation mice models. Both in the serum and BALF.

Response: Thank you for mentioning this. We received a similar comment from Reviewer #3 (first round of review from **Redacted**) and addressed it as follows. H&E staining of liver, lung and kidney tissues from this experiment showed reduced leukocyte infiltration after treatment with engineered therapeutic EVs, indicating organ protection (**Manuscript Fig. 7i,j and supplementary Fig. s16e**).

Fig. 7i,j: Liver and kidney protection by engineered EVs in LPS-induced mice model.

Fig. s16e: Lung protection by engineered EVs in LPS-induced mice model.

However, for PD biomarkers, including cytokines as suggested, obtaining significant data was challenging since only one or two mice survived in the control groups (LPS+PBS and LPS+CD63-Intein-SR).

7. what is the safety level when using high dose of EVs for the delivery of cargoes? Could the authors provide H&E staining of the main organs after high dosing of EVs?

Response: Thank you advising this. We have conducted H&E staining of the brain following osmotic minipump ICV injections (8×10^{11} EVs/mouse), a notably high EV dose, particularly

for localized brain injections. No significant inflammatory cell infiltration was observed in different brain regions for the engineered EV groups compared to the PBS group (**supplementary Fig. s14b**). Additionally, IHC staining of p65 in the hippocampus, cortex, and cerebellum showed no p65 overexpression post-injection (**supplementary Fig. s14c**), indicating that the engineered EVs exhibited minimal toxicity even at high doses.

Fig. s14b: Leukocyte infiltration in mice brains after osmotic minipump ICV injections of engineered EVs.

Fig. s14c: IHC staining of p65 in mice brains after osmotic minipump ICV injections of engineered EVs.

For other major organs, including the liver, kidney, and lung, H&E staining was performed to assess the protective effects of engineered EVs (5×10^{11} EVs/mouse) in an LPS-induced inflammation model (see the above data in response to your comment 6). In the liver, VSV-G-Foldon-Intein-SR EVs effectively reversed LPS-induced pathogenic effects, restoring

conditions similar to the PBS group without LPS induction. Inflammation levels in the kidney and lung were also reduced upon engineered EV treatment. These findings suggest that well-designed EVs not only avoid toxicity but also demonstrate protective effects.